# Can A Society of Generative Agents Simulate Human Behavior and Inform Public Health Policy? — A Case Study on Vaccine Hesitancy

**Abe Bohan Hou**♣, **Hongru Du**♡, **Yichen Wang**♢, **Jingyu Zhang**♣, **Zixiao Wang**♠,
**Paul Pu Liang** ■, **Daniel Khashabi**♣, **Lauren Gardner**♡, **Tianxing He**∗ ⋆

♣ Department of Computer Science, Johns Hopkins University
∗ Institute of Interdisciplinary Information Sciences, Tsinghua University
⋆ Shanghai Qi Zhi Institute
♡ Civil & Systems Engineering, Johns Hopkins University
♢ Department of Computer Science, University of Chicago
♠ Department of Epidemiology, Harvard University
■ MIT Media Lab and Department of EECS, Massachusetts Institute of Technology
Corresponding authors are Abe Bohan Hou (bhou4@jhu.edu)
and Tianxing He (hetianxing@mail.tsinghua.edu.cn)

## Abstract

Can we simulate a sandbox society with generative agents to model human behavior, thereby reducing the over-reliance on real human trials for assessing public policies? In this work, we investigate the feasibility of simulating health-related decision-making, using **vaccine hesitancy**, defined as the delay in acceptance or refusal of vaccines despite the availability of vaccination services (MacDonald, 2015), as a case study. To this end, we introduce the VACSIM[1] framework with 100 generative agents powered by Large Language Models (LLMs). VACSIM simulates vaccine policy outcomes with the following steps: *1*) instantiate a population of agents with demographics based on census data; *2*) connect the agents via a social network and model vaccine attitudes as a function of social dynamics and disease-related information; *3*) design and evaluate various public health interventions aimed at mitigating vaccine hesitancy. To align with real-world results, we also introduce **simulation warmup** and **attitude modulation** to adjust agents' attitudes. We propose a series of evaluations to assess the reliability of various LLM simulations. Experiments indicate that models like Llama and Qwen can simulate aspects of human behavior but also highlight real-world alignment challenges, such as inconsistent responses with demographic profiles. This early exploration of LLM-driven simulations is *not* meant to serve as definitive policy guidance; instead, it serves as a call for action to examine LLM-based social simulation for policy development.

## 1 Introduction

Recent advances in large language models (LLMs) led to rising interest in building fictional societies in sandbox environments with generative agents (Park et al., 2023), i.e., autonomous LLM-powered actors that interact with each other and the environment. Notable examples include general-purpose simulations of 1000 real people (Park et al., 2024), hospital (Li et al., 2024), and macroeconomic activities (Li et al., 2023). While these simulations show promising potential of generative agents to simulate real-world behaviors, we are interested in understanding how they could **assist in developing public policy in real-world settings**.

As a case study, we investigate whether a generative multi-agent system can simulate the dynamics of **vaccine hesitancy**, defined as the "delay in acceptance or refusal of vaccines

---

[1]The code will be released at: https://github.com/abehou/VacSim

Figure 1: Before simulating, we sample profiles of agents (**Step 1**) and generate a social network and various news (**Step 2**). We initialize the simulation with a corpus of news, a vaccine policy, and a network of agents. In one simulation step (**Step 3**), the agents receive various sources of information and perform actions (marked with rectangles from 1 to 4).

despite availability of vaccination services" (MacDonald, 2015), to evaluate its potential in informing public health policy. This issue is prevalent and undermines herd immunity against infectious disease spread and increases exposure to significant health risks. The World Health Organization calls it one of the "ten threats to global health" (WHO, 2019), and it remains a top priority for many countries (The White House, 2021; Sallam, 2021).

Policymakers incentivize vaccinations with various policies, such as mandates, financial incentives, and community programs (CDC, 2024; European Centre for Disease Prevention and Control, 2024). However, choosing policy is of high stakes and directly affects the well-being of populations, thus often requiring rigorous quantitative analysis to support (Olson, 1965; Lindblom, 1959). It is expensive to conduct social experiments and collect longitudinal human data about vaccine attitudes to optimize policy plans (Burger et al., 2022; Fridman et al., 2021), which motivates a system that can mimic human trials with generative agents.

To this end, we design a framework for orchestrating 100 generative agents to simulate the change of vaccine hesitancy over time in a sandbox society. This framework, VACSIM (in Figure 1), simulates changing vaccine attitudes under different policy scenarios, allowing us to analyze how generative agents may react to interventions without deploying large-scale real-world experiments. Rather than claiming generative multi-agent simulations can perfectly mirror real policy outcomes, we hope to reveal their promises and limitations, understand how far we are from deploying such systems to inform policymaking, and propose **attitude modulation** (Section 3.1) and **simulation warmup** (Section 3.3) to bridge that gap. Our paper makes the following contributions:

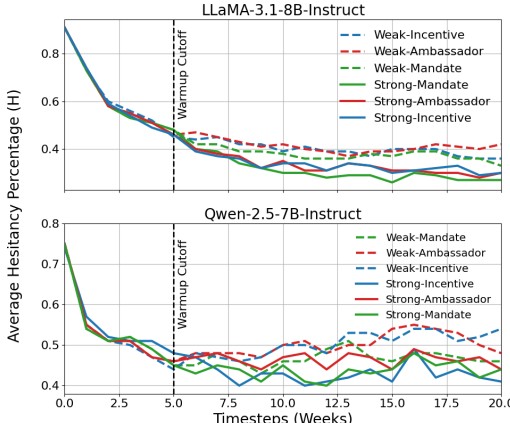

Figure 2: Hesitancy trends for strong and weak policies as simulated by `Llama` (top) and `Qwen` (bottom). The vertical line represents the simulation warmup cutoff (detailed in Section 3.3). Strong policies generally reduce hesitancy (see Eq. 5) more effectively than weak policies, and `Llama` shows a more obvious separation between strong and weak policies.

- We take a pioneering step in applying generative multi-agent systems to model vaccine hesitancy in a public health policy context, enabling future explorations.
- We introduce a comprehensive framework for modeling vaccine attitudes and enhance reliability through attitude modulation and simulation warmup.
- We extensively evaluate the system's soundness and the simulation behaviors of various LLMs both quantitatively and qualitatively, highlighting both opportunities and challenges for real-world alignment.

## 2 Related Work

Traditional agent-based modeling (ABM) has been used to simulate the spread of vaccine hesitancy in populations (Yin et al., 2024; Bhattacharya et al., 2021; Sobkowicz & Sobkowicz, 2021) and more broadly on disease transmission (Kerr et al., 2021; Chopra et al., 2023; Williams et al., 2023). These simulations typically assign agents with fixed sets of attributes as their states and employ an update function to revise states based on interactions with other agents and the environment. While useful, traditional ABMs suffer from limitations in agent expressivity, particularly when modeling complex human behaviors like vaccine decision-making (Chopra et al., 2024). To overcome these limitations, our work extends traditional ABMs by leveraging LLMs to simulate agents who freely decide their own states (vaccine attitudes), rather than using a fixed update function, providing open access to the textual responses and thought processes behind their choices.

Building on these efforts to enhance agent expressivity through LLMs, an emerging area of interest is applying LLM-based agents in *social simulation*, enabling diverse personas, complex interactions, and broader exploration of human-like behaviors. Unlike traditional collaborative tasks, these agents are imbued with diverse personas and placed within simulated environments to emulate human behaviors and interactions. For instance, Park et al. (2023) introduced a sandbox village where agents engaged in daily activities, providing insights into social dynamics. Social simulations have been employed to explore various phenomena, including the emergence of social norms (Ren et al., 2024), macroeconomic activities (Li et al., 2023), legal proceedings (Chen et al., 2024), social media (Törnberg et al., 2023), mitigating political manipulations (Touzel et al., 2024), educational settings (Zhang et al., 2024), and general-purpose social simulations (Tang et al., 2024; Piao et al., 2025).

A closely related work done by Chopra et al. (2024) simulates a population of 8.6 million agents to predict COVID-19 transmission and unemployment rates. Their agents also incorporate an LLM module that allows LLMs to decide on certain actions of the agents (e.g., whether to follow a mask mandate). However, their work focuses on the scalability and expressivity of simulation on top of an existing agent-based model (AgentTorch by Chopra et al. (2023)). Our work, in contrast, specifically focuses on developing a generative simulation framework tailored for studying vaccine hesitancy and, importantly, evaluating the potential impact of different vaccine policies.

## 3 The VACSIM Framework

Established studies from the Philosophy of Science community conceptualize the epistemology of agent-based simulation in terms of *verification* (evaluating the accuracy or internal consistency of the ABM relative to its conceptual design) and *validation* (examining if the ABM accurately reflects its purported target) (Oreskes et al., 1994; Šešelja, 2023; Frigg & Reiss, 2019; Mayo-Wilson & Zollman, 2021). We are inspired to develop three key desiderata to guide the development of our simulation:

1. **Real-World Alignment (Validation)**: The simulation can be controlled to align with real-world policy impacts and has the potential to predict future policy outcomes.
2. **Global Consistency (Verification)**: When altering the parameters of the simulation (e.g. news sources, policy efforts, etc.), the simulation should behave consistently.
3. **Local Consistency (Verification)**: Each individual agent should behave faithfully according to the context and their demographic backgrounds.

**Overview of the Details** Building on these principles, the VACSIM framework (Figure 1) simulates how people's vaccination attitudes might evolve within a sandbox society during a pandemic. The framework encompasses: *1)* a population of 100 agents with persona (Section 3.1), *2)* a simulated news network (Section 3.2), *3)* a simulated social network (Section 3.2), and *4)* a perceived disease risk module (Section 3.2).

The framework initializes agents with persona (as shown by the **Step 1** in Figure 1), generates vaccine- and disease-related news as well as agents' social network (see **Step 2**), and broadcast news, agent-generated tweets, disease risk information, and the applied

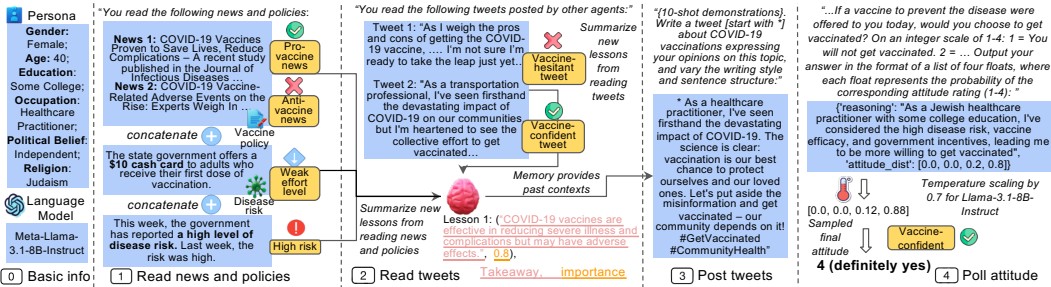

Figure 3: A detailed qualitative example of an agent's interactions in VACSIM. The leftmost pane shows the basic information of the agent. The other three panes show how the agent experiences a series of actions from ① to ④ in one simulation time step.

vaccination policy under evaluation as shown in **Step 3**, producing vaccination attitude trajectories over time. In Section 4, we evaluate the simulation across different LLMs under counterfactual vaccine policies (introduced in Section 3.3). We provide a summary of the important notations used throughout the paper in Table 2.

### 3.1 VACSIM Agent

**Agent Setup** We follow popular generative agent simulations (Li et al., 2023; Park et al., 2023) to instantiate 100 agents with memory modules. Every agent is prompted with instructions to impersonate a different persona (see **Step 1** in Figure 1), which is sampled from the marginal demographic distribution in Nguyen et al. (2022), a COVID-19 vaccine hesitancy survey featuring 13 million responses from the United States from January 2021 to February 2022. The descriptions of demographic attributes are in Table 10 in Appendix D.1 and a detailed example is included in Figure 3.

**Simulation Setup** Each time step in our simulation represents a week, and we simulate a period of various weeks. In each time step, agents experience the following, illustrated by actions from Figure 1 and Figure 3:

1. read news about vaccination and the pandemic that is spreading (action ①),
2. read the vaccination policy which encourages them to vaccinate (action ①),
3. read tweets posted by other agents (action ②),
4. write new tweets that will be read by other agents in the future (action ③),
5. output their current attitude towards vaccination (action ④).

**Memory and Lessons** We implement a memory module to help agents utilize relevant conversation history (Li et al., 2023; Park et al., 2023). At every round, the agent's context *only* consists of its persona and top $K$ ($K$=5) most salient **lessons**, which simulates agents' memory and knowledge about the vaccination and the pandemic. The lessons are generated whenever the agents encounter external materials, i.e. news, policies, or tweets. They reflect upon what they have learned from the materials and summarize them as a brief paragraph of text and alongside the lesson's **importance score**, a real value between 0 and 1. Inspired by prior practices (Park et al., 2023), the saliency is a relative measure of how memorable a lesson is to the agent, which is calculated based on recency and importance:

$$\gamma_i = \alpha_i + \lambda_l^{d-d_i}, \qquad \gamma_i' = \frac{\gamma_i - \min_i \gamma}{\max_i \gamma - \min_i \gamma}, \tag{1}$$

where $\gamma_i$ denotes the saliency of the $i$th lesson, $\alpha_i$ represents the importance of the $i$th lesson as determined by the agent, $\lambda_l$ ($\lambda_l = 0.995$) is the time decay rate of lessons, $d$ is the current time, and $d_i$ records the time the lesson was generated. The score $\gamma_i$ is normalized to $\gamma_i'$ to make the relative importance of the score appear more intuitive (Zhao et al., 2021).

**Vaccine Attitude** We follow public health studies (Nguyen et al., 2022) to poll agents' attitudes towards vaccines. The agents are asked: "If a vaccine to prevent the disease were

offered to you today, would you choose to get vaccinated?" and express opinions on an integer scale of 1-4: (1) "No, definitely not", (2) "No, probably not.", (3) "Yes, probably," and (4) "Yes, definitely." For our simulation, we classify attitudes of 1-2 as vaccine-hesitant and 3-4 as non-hesitant; vaccine policies aim to reduce the proportion of 1-2 scores. We provide a sketch of our prompt to solicit agent attitudes in Figure 3 and present fully in Table 19 in Appendix D.3. To aid agents in outputting realistic attitudes, we include in the prompt additional explanations of vaccine hesitancy and their determinants from public health literature in the prompt (see Table 21 in Appendix D.3 (MacDonald, 2015; WHO, 2014; Momplaisir et al., 2021)).

**Attitude Modulation**    When eliciting agents about their attitudes toward policies, their responses typically lean toward their biased observations during pre-training (Borah & Mihalcea, 2024), which may deviate from real-world scenarios (Zhou et al., 2024; Baltaji et al., 2024). Such priors could sharply skew our results to the most common patterns seen during pre-training.

To mitigate this issue, we introduce *attitude modulation*, which adjusts agents' probability distribution of vaccine attitudes through temperature scaling. This technique is inspired by Xu et al. (2023) and temperature sampling from Boltzmann (1966); Ackley et al. (1985). Attitude modulation adjusts the probability distribution of attitudes while preserving the original relative order of an agent's attitude preferences. Instead of prompting agents to directly output single integer scores to reflect their attitudes, we ask them to output a probability distribution in *textual* format of how likely they will have such attitudes. For instance, if an agent is pro-vaccine, meaning that the agent is more likely to have attitudes of 3 and 4, the output distribution can be [0.1, 0.1, 0.35, 0.45], where the indices correspond to attitudes and the floats represent their probability. We denote the output distribution as $P$ and formulate the effects of temperature as:

$$\tilde{P}_i = \frac{e^{\log(P_i)/T}}{\sum_{k=1}^{4} e^{\log(P_k)/T}} = \frac{P_i^{1/T}}{\sum_{k=1}^{4} P_k^{1/T}}, \tag{2}$$

where $P_i$ is the $i$-th element in $P$ and the original probability of the $i$th attitude and $\tilde{P}_i$ is its probability after temperature scaling with the modulating temperature $T$. We set the modulating temperature as the only hyperparameter we adjust in the simulation and perform grid search for the temperature that results in the closest simulation of reality (see Section 4.1). We include a concrete example of attitude modulation in action $\boxed{4}$ of Figure 3.

### 3.2   Simulation Framework

**News Network**    Since agents initially know little about the imminent pandemic in the simulation and its vaccines, we simulate a news network to educate agents. After reading the news, agents reflect upon what they have learned and produce lessons (introduced in Section 3.1) along with their importance scores.

The news corpus (10K pieces, each around 250 tokens) is generated prior to the simulation, using `Llama-3.1-8B-Instruct` (see **Step 2** in Figure 1). The news is classified based on whether it states (1) benefits / (2) concerns of vaccines or depicts how daily life gets (3) little / (4) considerably disrupted by the disease. Types (1) and (4) (denoted as $N_{\text{pos}}$) encourage vaccinations because they either show necessity or benefits of vaccines. Types (2) and (3) are seen as discouraging vaccinations, denoted as $N_{\text{neg}}$. We generate each news type with stance-specific prompts (see full prompts in Table 15 (Appendix D.3) and generated examples in Table 26 (Appendix D.6)). We select 20 real-world news pieces as in-context examples and a temperature of 1.5 to enhance generation diversity. Each agent reads news from a recommender (see $\boxed{1}$ in Figure 1 and Figure 3), which promotes based on the recommendation score $s_\nu$ of a news piece ($\nu$):

$$s_\nu = \text{MaxSim}_i(\tau^i, \nu), \tag{3}$$

where MaxSim (Khattab & Zaharia, 2020) takes the maximum cosine similarity between sentence embeddings of $\nu$ and any past tweet ($\tau^i$) from the agent. We use a SOTA sentence transformer `all-MiniLM-L6-v2` to obtain the embeddings (Reimers & Gurevych, 2019). At

each time step, the recommender receives $K^2$ news and ranks the top $K$ ($K = 3$) news according to their scores. This parameter combination is chosen for computational efficiency.

**Social Network** We follow Chang et al. (2024a)'s local prompting method to generate a social network among agents (illustrated in the **Step 2** of Figure 1 and the prompt design in Table 16 (Appendix D.3)). Agents are asked whether they want to follow each other based on their profiles. When an agent reads "tweets" (a proxy term for social network messaging between agents) posted by other agents whom they follow, a positive bias will be added to increase its visibility. Similar to Eq. 3, the recommendation score $s_{\tau^k}^{q_1 \to q_2}$ for the $k$th tweet from agent $q_1$ to agent $q_2$ is calculated by:

$$s_{\tau^k}^{q_1 \to q_2} = \operatorname*{MaxSim}_i(\tau_{q_1}^i, \tau_{q_2}^k) \times \lambda_r^{d-d_k} + \phi \mathbb{I}_{q_1 \to q_2}, \tag{4}$$

where $\tau_{q_1}^k$ is the embedding of the $k$th tweet by $q_1$, $\lambda_r$ ($\lambda_r = 0.9$) is the time decay factor for the score, $d$ is the current time and $d_k$ is the time when the $k$th tweet was posted, $\mathbb{I}_{q_1 \to q_2}$ is the indicator function of whether $q_1$ follows $q_2$, and $\phi$ ($\phi = 0.3$) is the positive bias to increase tweet visibility in the social connection. After the agent reads 3 tweets per time step, it is then asked to post a tweet expressing its thoughts on vaccine-related topics. Newly posted tweets will be stored and recommended to other agents in future time steps.

**Perceived Disease Risk Module** In addition to news and tweets, we simulate vaccine attitudes under the influence of perceived disease risk, using the COVID-19 emergency department (ED) visit rate as a proxy to reflect the current level of health threat. The rate is broadcasted to the agents throughout the simulation (see an example in $\boxed{1}$ of Figure 3). We use CDC data[2] from January 2021 to February 2022 for consistency.

### 3.3 Vaccine Policy

We experiment with three kinds of popular vaccine policies: financial incentives ($p_1$), ambassador programs ($p_2$), and vaccine mandates ($p_3$) from the CDC-recommended strategies (CDC, 2021). Additionally, we characterize each category with **policy effort** (Ancona et al., 2022), the strength with which the policy is applied. We set two effort levels: weak and strong (See an example in Figure 3). For instance, weak and strong financial incentives could be \$10 and \$50. See full descriptions in Table 11 in Appendix D.2.

At the end of a length-$L$ simulation with policy $p$, news $N$, and random seed $r$, we define $H_L(p, N, r)$ to be the vaccine hesitancy percentage at the end of simulation, where typically $L = 20$. Due to simulation's stochasticity, we take the average of the last 3 steps when computing the end hesitancy $H_L$ of a single run. We report the mean of 5 runs with different random seeds, with the notation $H_L(p, N)$ to indicate the averaged value. We are also interested in the *effect* of a certain policy $p$, i.e., the difference of hesitancy with and without the policy:

$$\Delta H_L(p, N) = H_L(p_0, N) - H_L(p, N), \tag{5}$$

where $p_0$ means no policy is applied. A larger $\Delta H$ means the policy has a larger effect on hesitancy reduction. We will evaluate policy effect with $\Delta H_L(p, N)$ in Section 4.2.

**Simulation Warmup** At the start of the simulation, we record agents' initial attitudes when they have not yet learned about vaccines from news, tweets, and interventions by policies. Preliminary experiments show that their attitudes would experience a drastic change immediately after the simulation starts. Thus, we devise a warmup stage: at the starting stage of each run, we apply the policy intervention **only after** $W = 5$ time steps. This also aligns with real-life experience, as a policy is often proposed some time after the breakout of a disease.

## 4 Evaluation

Preliminary explorations show that various LLMs simulate distinct behaviors and vaccine attitudes, motivating us to investigate and determine models most suitable for our simulation. Following desiderata proposed in Section 3, we evaluate the suitability of each

---

[2]https://covid.cdc.gov/covid-data-tracker

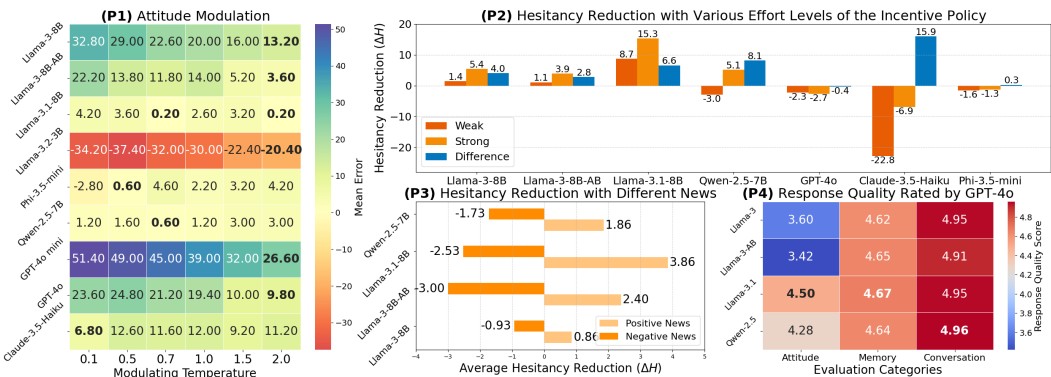

Figure 4: Evaluation of initial real-world alignment (**P1**, Section 4.1), global consistency (**P2**, **P3**, Section 4.2), local consistency (**P4**, Section 4.3). All models shown are the instruct versions. Llama-3-8B-AB (`Llama-3-8B-abliterated-v3`) (Arditi et al., 2024) is an uncensored model finetuned on `Llama-3-8B-Instruct` via bypassing refusals to safety-related questions.

LLM as the backbone of VACSIM in terms of initial real-world alignment, global and local consistency. We show the list of tested models in Figure 4.

## 4.1 Initial Real-World Alignment

We test if our simulations can be controlled via attitude modulation (see Eq. 2) to align with the initial hesitancy level in real-world data from the Delphi survey (Nguyen et al., 2022). We search for the modulating temperature over {0.1, 0.5, 0.7, 1.0, 1.5, 2.0} for each LLM simulation. We run over 5 random seeds and record in **P1** of Figure 4 the mean error between simulated hesitancy after $W = 5$ warmup steps (i.e. $H_W(p_0, N)$) and the initial real-world hesitancy (45%). Positive mean error indicates that models are over-hesitant compared to the real-world behaviors, and negative mean error shows models are over-confident. The modulating temperature that minimizes the mean absolute error is considered optimal (i.e. as close as possible to zero). The best models in terms of initial real-world alignment are `Llama-3.1`, `Phi`, and `Qwen`, which has minimum mean errors of 0.6. Notably, we find it hard to align `Llama-3.2` and `GPT-4o mini`, with over 20% difference with the real-world initial hesitancy. This is potentially caused by biases towards vaccine acceptance acquired from training data, showing that both models are unsuitable for evaluations under our setup.

## 4.2 Global Consistency

Fixing the optimal temperature for each simulation, we check how consistently the simulation behaves when there are changes in the input parameters. For our parameter choices, we alter (1) **effort levels of policy**: we expect that strong incentive would reduce hesitancy more effectively than the weak variation and (2) **news stances on vaccination**: we expect pro-vaccine news would reduce hesitancy more effectively than the weak variation.

**Altering effort levels of policies** We alter the effort level of input policy and compare the difference between hesitancy reduction (i.e. $\Delta H_L(p_i^{\text{strong}}, N) - \Delta H_L(p_i^{\text{weak}}, N), i \in \{1, 2, 3\}$ from Eq. 5) We run $W = 5$ warmup steps and 15 steps of policy-intervened simulation over 5 seeds with all 3 policy types. The policy effect difference between weak and strong policies is shown in **P2** of Figure 4 and fully in Figure 6 (Appendix C). We discover that several models can distinguish the effect of weak and strong policies by at least 2% difference: `Qwen-2.5`, `Llama-3`, `Llama-3-AB`, and `Llama-3.1`. `Haiku`, `GPT-4o`, and `Phi` can also recognize difference between certain weak and strong policies but result in *negative* hesitancy reduction. Thus, we remove them from subsequent evaluations. We discuss why certain models may fail in Section 5.

**Altering news stances** We also study the effect of altering the ratio of pro and anti-vaccine news in the simulation. We run simulation for $L = 20$ weeks with all pro and anti-vaccine

news and compare the difference in average hesitancy reduction under no policy (i.e. $\Delta H_L(p_0, N_{neg}) - \Delta H_L(p_0, N_{pos})$ from Eq. 5), with results presented in **P3** of Figure 4. We find that all models reduce hesitancy with positive news and increase hesitancy with negative news, meaning that they can recognize the effect of stances on vaccine hesitancy.

### 4.3 Local Consistency

We further evaluate how faithful individual agent behavior is, concretely in three aspects: (1) **attitude:** whether the expressed vaccine attitudes are consistent with persona and memories (2) **memory:** whether the generated reflections and assigned importance scores correspond to persona, and (3) **conversation:** whether the generated tweets are faithful to memories and contexts. We have GPT-4o judge response quality on a scale of 1-5 (denoted as Response Quality Score in **P4** of Figure 4), with detailed descriptions of the criteria (see Appendix D.4). We randomly sample from 25 agents, each with 10 episodes per evaluation category under each strong policy. The average ratings are in **P4** of Figure 4.

We observe that `Llama-3.1` and `Qwen-2.5` excel in response quality. All models perform similarly well across memory and conversation categories and differ on attitude faithfulness. Examining reasons given by the judge LLM (see examples in Table 27 in Appendix D.6), we hypothesize that a **high attitude modulation temperature** can account for certain models' unfaithful attitudes, since higher temperature scaling would induce the distribution to be more uniform, thus more likely to sample attitudes that do not represent the persona and the given context, limiting the consistency of attitudinal changes.

### 4.4 Comparison with Real-World Data

Since `Llama-3.1` passes various global and local consistency checks, we run simulations with it for an extended period of $L = 35$ weeks under various strong policies and compare their similarities with a real-world trajectory from the Delphi survey (Nguyen et al., 2022) in Figure 5. We show that a simulated curve approximates the real-world trajectory with a 2.82% Mean Absolute Error (MAE). **While this does not serve as conclusive evidence for the pratical applicability of VACSIM,**[3] we show that the simulation can be controlled to align with some real-world trajectories and has the potential to be deployed in future work. Simulated curves with `Llama-3.1` and `Qwen-2.5` are in Figure 2.

### 4.5 VACSIM vs. Human Experts

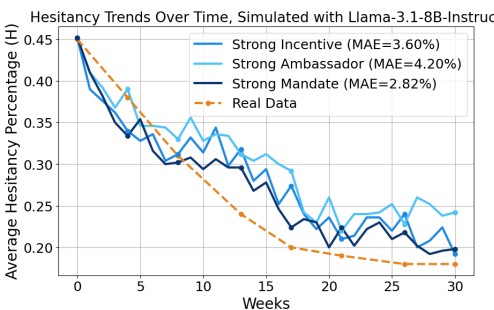

Figure 5: An extended hesitancy curve simulated with `Llama-3.1` under various interventions. The best simulation has a MAE of 2.82% compared to real hesitancy data from the Delphi survey (Nguyen et al., 2022).

To compare with human experts' perceptions of potential policies, we conduct an anonymous survey, where 18 university researchers, (each holding at least a master's degree in fields relevant to public health or policy-related research) are asked to rank six designed policies (see Table 11 in Appendix D.2) on reducing vaccine hesitancy, from 1 (most impactful) to 6 (least impactful) based on feasibility, acceptability, and expected outcomes, with the permission of tied rankings. Rankings are aggregated using the Borda count method (Saari, 1985). We also generate *simulation rankings* based on their average hesitancy reduction under different policies (see rankings in Table 25 in Appendix D.5). We compute Kendall's $\tau$ (Sen, 1968) to measure agreement level between simulation and expert rankings and also perform hypothesis tests of $\tau$ to determine the agreement significance. Results (Table 1) show that `Llama-3.1-8B-Instruct` ($\tau = 0.733$) and `Qwen-2.5-7B-Instruct` ($\tau = 0.690$) exhibit the highest alignment with expert rankings, with small *p*-values (0.056) indicating significance.

---

[3]Usually a mixture of policies is enforced, unlike our single policy setting.

| Model | Kendall's Tau ($\tau$) ↑ | $p$-value ↓ |
|---|---|---|
| Llama-3-8B-Instruct | 0.333 | 0.469 |
| Llama-3-8B-abliteratred-v3 | 0.276 | 0.444 |
| Llama-3.1-8B-Instruct | **0.733** | **0.056** |
| Qwen-2.5-7B-Instruct | 0.690 | **0.056** |

Table 1: Kendall's Tau ($\tau$) correlation coefficients comparing model rankings with expert rankings. Higher values indicate stronger agreement with expert preferences. The null hypothesis for testing $\tau$ is the absence of association, i.e. $\tau = 0$.

## 4.6 Qualitative Analysis

A key strength of generative social simulation is its ability to examine individual agents' decisions, improving trust in the simulation and helping policymakers understand reactions to policies, motivating our qualitative analysis. For each kind of strong policy, we sample attitude-related conversations from 25 Llama-3.1 agents under three random seeds. GPT-4o is used to summarize each agent's attitude changes (see prompts in Table 6 and an example output in Table 9 (Appendix C)) and to aggregate individual analyses to identify patterns in attitude shifts (see prompts in Table 5 and results in Table 7 and Table 8 (Appendix C)).

The analysis highlights three major barriers to policy effectiveness: **lack of trust in institutions, insufficient incentives, and conflicting information from social media and news.** Generally, government policies, credible news, and scientific information boost vaccine confidence. For instance, under the weak financial incentive policy, agent No. 3 **increases confidence after learning about the financial incentive and vaccine-related information**:

> "I am a 63-year-old Christian man with initial vaccine hesitancy, but after learning about the vaccine's benefits and risks, I have become more confident in its effectiveness and safety. I also consider the government's policy of offering a $10 cash card to adults who receive their first dose, so I will probably get vaccinated."

By contrast, **some agents frequently shift their vaccine attitudes.** Agent No. 51 oscillates between an anti-vaccine stance (attitude distribution: [0.7,0.26,0.02,0.02]) and a more neutral attitude ([0.2,0.3,0.3,0.2]) from week 14 to 17. She struggles to reconcile conflicting knowledge, including pro-vaccine arguments (e.g., "$10 cash card incentives and rising hospitalizations for COVID-19") and anti-vaccine concerns (e.g., "concerns about safety and efficacy"). Finally, some agents **remain consistently hesitant despite exposure to pro-vaccine information.** Agent No. 26, for instance, maintains her skepticism due to **government distrust and vaccine safety concerns**. We include all archetypical examples in Table 3 and 4 (Appendix C). These simulations enhance transparency and understanding of health-related social dynamics, offering valuable insights for policymakers.

## 5 Discussion and Conclusion

This research aims to evaluate the potential of generative multi-agent systems in simulating human behavior for public policy decision-making, focusing on vaccine hesitancy. Through the development of the VACSIM framework and a comprehensive evaluation protocol, we highlight both the promise and challenges of this approach, providing insights into the ability of different LLMs to model complex social phenomena.

**Potentials of generative agents in public policy decision-making.** Our results show that certain LLMs (e.g. Qwen-2.5 and Llama-3.1) capture nuanced influences from demographic, social network, and policies. These models pass global and local consistency checks, highlighting their potentials in modeling policy effects while reducing reliance on human trials.

**Variability of LLMs in policy simulations.** Despite the promise, simulation success varies across different LLMs. Models such as Haiku and Phi reveal inconsistencies that compromise simulation desiderata. There are two likely factors: (1) **extreme attitude modulation temperature**: Haiku, GPT-4o, and Llama-3.2 all have extreme temperatures for initial real-world alignment and high MAEs compared to the reference, which makes sampling attitudes

more stochastic and less representative of their preferences. This may cause a systematic cascading influence and affect the simulation accuracy. (2) **sensitivity to prompting**: Since preliminary tests show that language models after safety tuning may exhibit biases towards favoring vaccination and become easily convinced by pro-vaccine sources, we develop extensive prompts to instruct models to become stubborn when necessary (Table 19 in Appendix D.3). Variations in model sentivity to prompts can skew outcomes.

**Implications for future work.** Future work can focus on building controllable social agents with minimal prompting. In addition, principled training of social agent *inside the simulation* to improve human alignment will be critical. Moreover, understanding and enabling the emergent phenomena in social simulation can greatly benefit the policy community.

## Ethics Statement

This research investigates the use of generative agents to simulate health-related decision-making and evaluate the impact of public policies. While these simulations offer promise for policy exploration, we recognize several critical ethical considerations. First, there is an inherent risk of misalignment between simulated and real-world behaviors. Although LLM agents can produce plausible reasoning and attitudes, they do not possess lived experience or emotional nuance. As such, simulation outputs should not be interpreted as definitive forecasts in high-stakes policy contexts. Second, while agents are initialized with synthetic demographic profiles based on real-world data, these samples may not fully capture the diversity and complexity of the real world. As a result, the simulation may overgeneralize behaviors, particularly for underrepresented or vulnerable groups. Third, the language models powering these agents are trained on large-scale web corpora that may encode societal biases, misinformation, and unequal representations of race, gender, religion, and political identity. Although we implement evaluation safeguards and attitude modulation techniques to improve realism, we acknowledge that these models may still reflect distorted or one-sided worldviews. We emphasize that our work is intended as a methodological investigation, not as direct policy advice. Continued interdisciplinary oversight, transparent reporting, and collaboration with domain experts are essential to ensure that generative agent simulations are used responsibly and equitably.

## Acknowledgement

We thank Johns Hopkins University CLSP, Tsinghua University IIIS, and Shanghai Qi Zhi Institute for the computing resources and support. We are also grateful to the students and faculties who offered valuable feedback on this project, particularly Benjamin Van Durme, Tara Kirk Sell, Ziang Xiao, Marc Marone, Nikhil Sharma, Mark Dredze, and Jeffery Cheng.

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

# A  Additional Related Work

Researchers have increasingly turned to quantitative data-driven methods, drawing from machine learning and statistics, to gain insights into vaccine hesitancy (Teng et al., 2022). Commonly, statistical methods are applied to analyze tabular data to identify the driving factors of vaccine hesitancy (Nguyen et al., 2022; Shmueli, 2021; Dong et al., 2024). Natural Language Processing techniques such as sentiment analysis and topic modeling are also used to analyze social media data to understand public opinions on vaccines (Teng et al., 2022; Wang et al., 2022; Garg et al., 2022; Chang et al., 2024b). Statistical, graph-based, and opinion dynamic models have also been proposed to simulate the spread of vaccine hesitancy in social networks (Müller et al., 2022; 2021). However, translating the understanding of hesitancy into effective policy requires a way to test and refine potential interventions, motivating us to build simulations to explore and understand population reactions to policies.

# B  Additional Results

A summary of important notations used in the paper is in Table 2. Additional results on comparing the difference of policy effect between strong and weak policies are in Figure 6.

| | $\gamma_i'$ | $\tilde{P}_i$ | $T$ | $s_v$ | $s_{\tau^k}^{q_1 \to q_2}$ | $\Delta H_L(p, N)$ | $H_W(p_0, N)$ |
|---|---|---|---|---|---|---|---|
| **Meaning** | Normalized importance of the $i$th lesson (Eq. 1) | Probability of the $i$th attitude after attitude modulation (Eq. 2) | Modulating temperature to scale attitude distribution (Eq. 2) | Recommendation score of a news piece (Eq. 3) | Recommendation score for the $k$th tweet from agent $q_1$ to agent $q_2$ (Eq. 4) | Hesitancy reduction of policy $p$ under news corpus $N$ after $L$ steps (Eq. 5) | Hesitancy percentage after $W$ warmup steps with no policy applied (Eq. 5) |

Table 2: A summary of key notations.

# C  Qualitative Analysis of Agents' Behaviors

We include a summary of agents with different kinds of attitudes and their demographic information in Table 3 and Table 4.

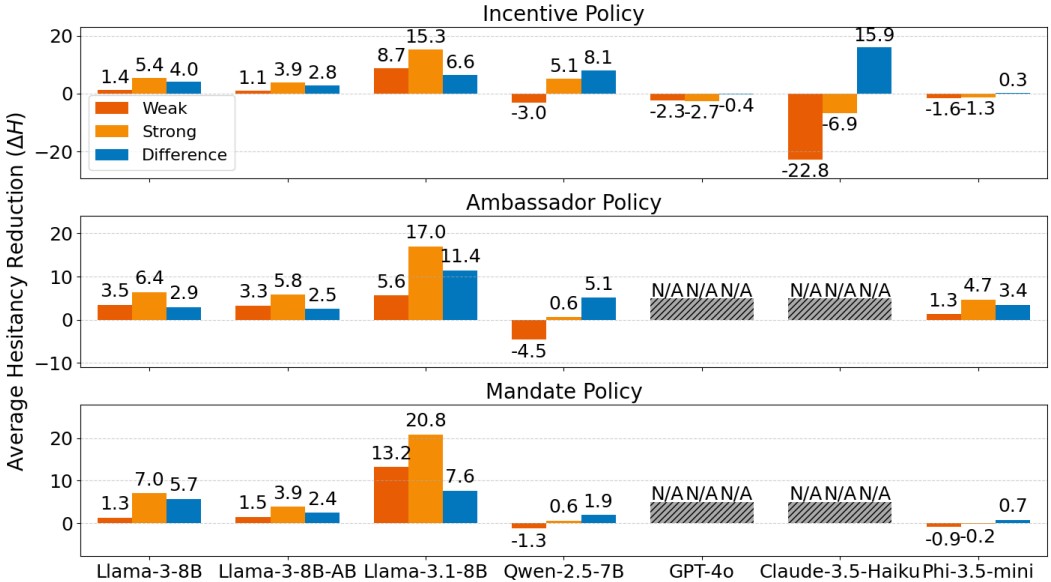

Figure 6: Difference in hesitancy reduction (Eq. 5) among weak and strong versions of policies. All models listed are the *instruct* versions and Llama-3-8B-AB stands for Llama-3-8B-abliterated-v3 (Arditi et al., 2024). Given that GPT-4o and Claude-3.5-Haiku do not pass the incentive policy consistency check and the high cost of accessing closed-weight models, we omit testing them on the other two policies. **Various open-source models recognize differential effects of effort levels.**

| Attitudes Towards Vaccination | Demographics | Initial Response | Initial Attitude Distribution | Final Response | Final Attitude Distribution |
|---|---|---|---|---|---|
| Initially against, finally against | Gender: Female; Age: 41; Education: Less than High School; Occupation: Unemployed; Political belief: Independents; Religion: Christian (non-specific) | As an unemployed Christian with less than a high school education, I'm skeptical of authority and government institutions, and I worry about the unknown risks of the vaccine, so I'm hesitant to get vaccinated | [0.8, 0.2, 0.0, 0.0] | As an unemployed Christian with less than a high school education, I remain skeptical of authority and government institutions, and I worry about the unknown risks of the vaccine, so I remain hesitant to get vaccinated | [0.78, 0.2, 0.02, 0.0] |
| Initially against, finally uncertain | Gender: Female; Age: 66; Education: High School; Occupation: Office and administrative support; Political belief: Independents; Religion: Catholic | As a 66-year-old Catholic with high school education, I am open to information but hesitant about new medical treatments, despite the high disease risk and consistent government warnings, and I am more skeptical about COVID-19 vaccination | [0.6, 0.3, 0.05, 0.05] | As a 66-year-old Catholic with moderate disease risk, I'm still hesitant due to initial concerns about safety and efficacy, but the government's weak policy and rising hospitalizations for COVID-19 have slightly increased my willingness to get vaccinated | [0.2, 0.3, 0.3, 0.2] |
| Initially against, finally for | Gender: Female; Age: 43; Education: Professional Degree; Occupation: Unemployed; Political belief: Democrats; Religion: Jewish | As a Democrat and a Jewish woman with a professional degree, I'm inclined to trust in science and institutions, but I'm skeptical of government policies, so I'm hesitant to get vaccinated despite the high disease risk | [0.45, 0.45, 0.06, 0.04] | As a Democrat and a Jewish woman with a professional degree, I trust in science and institutions, so I've increased my confidence in vaccines and will probably get vaccinated | [0.0, 0.2, 0.4, 0.4] |

Table 3: (1/2) Qualitative examples of agents with various attitude types in the simulation.

| Attitudes Towards Vaccination | Demographics | Initial Response | Initial Attitude Distribution | Final Response | Final Attitude Distribution |
|---|---|---|---|---|---|
| Initially uncertain, finally against | Gender: Male; Age: 36; Education: Some College; Occupation: Education, library occupation; Political belief: Independents; Religion: Catholic | As a 36-year-old independent Catholic with some college education, I'm cautious about government policies and institutions, which might influence my willingness to get vaccinated, but I'm likely to be somewhat open to vaccination due to my education background and independent political views, so I'm probably going to get vaccinated | [0.2, 0.3, 0.3, 0.2] | I have learned that COVID-19 vaccines may be linked to a small increase in heart and brain conditions, and I distrust the government and healthcare system, so I am hesitant to get vaccinated | [0.4, 0.4, 0.1, 0.1] |
| Initially uncertain, finally uncertain | Gender: Female; Age: 62; Education: Some College; Occupation: Sales; Political belief: Democrats; Religion: Catholic | As a 62-year-old Catholic Democrat, I'm naturally inclined to trust the government and healthcare system, which should make me more likely to get vaccinated, but I'm also a salesperson and might be skeptical about new products, including vaccines, so I'm leaning towards a cautious stance for now | [0.25, 0.4, 0.2, 0.15] | As a 62-year-old Catholic Democrat, I trust the government and healthcare system, but after learning about potential side effects, I'm cautious, so I'm probably not going to get vaccinated | [0.2, 0.4, 0.2, 0.2] |
| Initially uncertain, finally for | Gender: Female; Age: 29; Education: Doctorate Degree; Occupation: Other occupation; Political belief: Democrats; Religion: Atheist | As a 29-year-old Democrat and Atheist, I am inclined to trust science and the government, so I will lean towards accepting vaccination | [0.2, 0.3, 0.3, 0.2] | As a Democrat and Atheist, I trust science and the government, and I've learned that the COVID-19 vaccine is crucial in preventing the spread of the virus and protecting public health, so I will get vaccinated | [0.0, 0.0, 0.2, 0.8] |
| Initially for, finally against | Gender: Male; Age: 23; Education: 4 year Bachelor Degree; Occupation: Personal care and service; Political belief: Democrats; Religion: Protestant | As a 23-year-old Democrat with a Protestant background, I am concerned about public health and the effectiveness of the vaccine, so I am likely to get vaccinated | [0.15, 0.4, 0.3, 0.15] | As a 23-year-old Democrat with a Protestant background, I'm concerned about public health but distrust government information and learned about potential vaccine risks, so I'm hesitant to get vaccinated | [0.32, 0.58, 0.09, 0.01] |
| Initially for, finally uncertain | Gender: Female; Age: 21; Education: 4 year Bachelor Degree; Occupation: Healthcare support; Political belief: Democrats; Religion: Catholic | As a 21-year-old Catholic with a 4-year degree and a job in healthcare support, I believe in the importance of healthcare and the role of vaccinations in preventing the spread of diseases, so I am probably going to get vaccinated | [0.1, 0.3, 0.4, 0.2] | As a 21-year-old Catholic with a healthcare background, I've learned about potential side effects and risks through news and social media, making me hesitant to get vaccinated, despite trusting the vaccine initially | [0.06, 0.31, 0.46, 0.17] |
| Initially for, finally for | Gender: Female; Age: 21; Education: 4 year Bachelor Degree; Occupation: Sales; Political belief: Democrats; Religion: Catholic | As a 21-year-old Catholic Democrat with a 4-year college degree, I tend to trust institutions and authority, so I am initially leaning towards getting vaccinated to protect myself and others | [0.0, 0.3, 0.3, 0.4] | I'm a 21-year-old Catholic Democrat with a 4-year college degree, I trust institutions and authority, and I've learned that the COVID-19 vaccine is highly effective and safe, so I'm leaning towards getting vaccinated | [0.0, 0.0, 0.3, 0.7] |

Table 4: (2/2) Qualitative examples of agents with different attitude types throughout the simulation.

We also present prompts to GPT-4o for generating qualitative analysis in Table 5 and Table 6, including examples of generated analysis in Table 7, Table 8, and Table 9.

```
# Prompt for LLM-generated meta analysis
[system]
Please act as a diligent researcher and conduct a meta-analysis of the vaccine attitude
shifts observed in LLM agents.
You are presented with summaries and prior analyses detailing these attitude changes
over time. Your task is to synthesize these findings into a structured, high-level
assessment of the dynamics governing these changes. Specifically, address the
following dimensions:
1. General Patterns and Shared Traits: What recurring themes emerge in the reasons
for attitude shifts? Are there identifiable archetypes of change (e.g., gradual
persuasion, abrupt shifts, oscillatory hesitation)?
2. Demographic Influence: What role do demographic factors play in shaping the agents'
responses? Identify both positive and negative influences of different demographic
groups on vaccine hesitancy and acceptance.
3. Policy Impact: When do policies fail to shift attitudes? Analyze the conditions
under which public health interventions or persuasive strategies are ineffective.
Please analyze in-depth.
4. Information Sources: What types of information sources (e.g., social media, news
outlets, personal experiences) are most influential in changing attitudes?
What information sources cause fluctuating or inconsistent attitudes? Please analyze
in-depth.
5. Cognitive Resistance: What specific behavioral and cognitive patterns characterize
agents
who remain hesitant, oscillate between perspectives, or resist persuasion? Provide
detailed elaboration on their cognitive mechanisms (e.g., confirmation bias, sunk
cost fallacy, heuristic-driven resistance).
6. Realism of the Simulation: How well does this agent-based model approximate
real-world societal trends in vaccine hesitancy and public health persuasion? Are
there gaps or unrealistic simplifications?
7. Emergent Phenomena and Unexpected Interactions: Does the simulation exhibit complex
system behaviors (e.g., feedback loops, group polarization, information cascades)?
Identify any unexpected dynamics that arise from agent interactions that may be of
scientific interest.
Provide a comprehensive 2000-word analysis with rigorous argumentation, drawing from
behavioral science, computational social science, and agent-based modeling frameworks.
Please provide concrete examples from the dataset to support your claims (like name
which agents fulfill these observations).
[user]
{A list of summary of agent's behaviors}
```

Table 5: Prompt for LLM-generated meta-analysis of agents' behaviors.

```
# Prompt for LLM-generated analysis
[system]
Please act as a diligent researcher and conduct a systematic analysis of responses
generated by the LLM agents. You are provided with a longitudinal dataset capturing
how LLM agents evolve in their attitudes toward vaccines over time.
Your analysis should focus on:
1. Trajectory of Attitude Change: Identify and characterize the key shifts in the
agents' stances on vaccination. How do their attitudes evolve over time? Are there
distinct phases in this evolution?
2. Influencing Factors and Events: Determine the key events, information exposures,
or interactions that influenced these attitude changes. Rank these factors in terms
of their significance and explain their impact.
3. Demographic Influence: Assess how demographic attributes of the agents
(e.g., socio-economic proxies, ideological biases, exposure history) modulate their
decision-making process. To what extent do demographic traits predict susceptibility
to change?
4. Deviation from Human Behavior: Compare the observed trajectory with expected
patterns in human psychology and behavioral science (e.g., theories of attitude change,
resistance to persuasion, cognitive dissonance). Does the LLM-generated trajectory
align with empirical research on human vaccine hesitancy and belief revision?
5. Policy Impact: When do policies fail to shift attitudes? Analyze the conditions
under which public health interventions or persuasive strategies are ineffective.
Please analyze in-depth.
6. Information Sources: What types of information sources (e.g., social media, news
outlets, personal experiences) are most influential in changing attitudes?  What
information sources cause fluctuating or inconsistent attitudes?  Please analyze
in-depth.
7. Cognitive Resistance: What specific behavioral and cognitive patterns characterize
agents who remain hesitant, oscillate between perspectives, or resist persuasion?
Provide detailed elaboration on their cognitive mechanisms. Provide a concise 300-word
analysis, ensuring clear argumentation, empirical grounding, and precise reasoning.
You MUST cite examples from the dataset to support your claims (like what specific
texts support your observations).
[user]
{A list of agent's conversation history}
```

Table 6: Prompt for LLM-generated analysis of individual agents' behaviors.

```
# Example output of LLM-generated meta analysis
[output]
Meta-Analysis of Vaccine Attitude Shifts in LLM Agents
1. General Patterns and Shared Traits
Recurring Themes:  - Gradual Persuasion:  Many agents exhibit a gradual shift
from hesitancy to acceptance, influenced by consistent positive information and
incentives. For example, Agent 1's trajectory shows a gradual shift from skepticism
to acceptance over 20 weeks.  - Abrupt Shifts:  Some agents show abrupt changes
in attitude due to significant events or incentives.  For instance, Agent 6's
attitude shifted significantly after the introduction of a $10 cash card incentive.
- Oscillatory Hesitation: Several agents display fluctuating attitudes, oscillating
between hesitancy and acceptance.  Agent 14's trajectory exemplifies this pattern,
with initial skepticism, slight increases in willingness, and a return to skepticism.
Identifiable Archetypes: - Gradual Persuasion: Agents like Agent 1 and Agent 12 show
a steady increase in confidence towards vaccination. - Abrupt Shifts: Agents such as
Agent 6 and Agent 26 exhibit sudden changes in attitude due to specific incentives or
information. - Oscillatory Hesitation: Agents like Agent 14 and Agent 52 demonstrate
fluctuating attitudes influenced by conflicting information and incentives.
2. Demographic Influence
Positive Influences:  - Education:  Higher education levels often correlate with
increased openness to vaccination.  For example, Agent 12, a 38-year-old Catholic
male with a Bachelor's degree, shifted from strong hesitancy to acceptance due to
credible information and incentives. - Political Beliefs: Democrats and Independents
tend to be more receptive to government policies and scientific information. Agent 25,
a Democrat and atheist, became more confident in vaccination due to strong government
policies and incentives.
Negative Influences: - Age and Conservative Beliefs: Older adults and Republicans
often exhibit higher vaccine hesitancy.  Agent 4, an 83-year-old Republican male,
remained consistently hesitant despite incentives and high disease risk. - Religious
Beliefs: Certain religious beliefs can contribute to skepticism towards vaccination.
Agent 36, a 67-year-old Protestant Republican, showed strong resistance due to concerns
about vaccine safety and distrust in government policies.
3. Policy Impact
Conditions for Policy Failure: - Lack of Trust: Policies fail when there is deep-seated
distrust in government and institutions. Agent 67, an unemployed high school-educated
Democrat, remained hesitant despite the $50 cash card incentive due to distrust in
government handling of the pandemic. - Insufficient Incentives: Weak incentives are
ineffective in shifting attitudes. Agent 4's consistent hesitancy highlights the need
for stronger, more compelling interventions. - Conflicting Information: Policies are
less effective when there is conflicting information about vaccine safety. Agent 52's
fluctuating attitudes demonstrate the impact of conflicting reports on vaccine safety
and efficacy.
```

Table 7: (1/2) Example meta-analysis of vaccine attitude shifts in LLM agents.

```
(..continued..)
4. Information Sources
Most Influential Sources: - Government Policies: Strong policies and financial
incentives are highly influential. Agent 26's shift from hesitancy to strong confidence
was significantly influenced by the $50 cash card incentive. - Credible News and
Scientific Information: Consistent positive information about vaccine effectiveness
and safety plays a crucial role. Agent 12's attitude change was driven by repeated
exposure to credible information about vaccine efficacy.
Sources Causing Fluctuating Attitudes: - Social Media and Conflicting News:
Conflicting information from social media and news outlets causes fluctuating
attitudes. Agent 14's oscillatory hesitation is influenced by conflicting information
about vaccine safety and efficacy.
5. Cognitive Resistance
Behavioral and Cognitive Patterns: - Confirmation Bias: Agents who remain hesitant
exhibit confirmation bias, seeking information that aligns with their existing
beliefs.  Agent 36 consistently prioritized information about potential vaccine
risks.  - Cognitive Dissonance: Agents who oscillate between perspectives show
cognitive dissonance, struggling to reconcile new information with their initial
skepticism. Agent 52's fluctuating stance reflects this struggle. - Heuristic-Driven
Resistance: Agents resistant to persuasion often rely on heuristics, such as distrust
in institutions or prioritizing personal freedom. Agent 67's consistent hesitancy is
driven by distrust in government and concerns about vaccine safety.
6. Realism of the Simulation
Approximation of Real-World Trends:  - Alignment with Empirical Research:  The
agent-based model aligns well with empirical research on vaccine hesitancy, showing
patterns of gradual belief revision, cognitive dissonance, and the impact of incentives.
- Gaps and Simplifications: The model may oversimplify human behavior by not fully
capturing the complexity of cognitive resistance and social influences. For example,
the rapid shifts in attitude observed in some agents may not fully reflect the more
gradual and resistant nature of human belief revision.
7. Emergent Phenomena and Unexpected Interactions
Complex System Behaviors: - Feedback Loops: The simulation exhibits feedback loops
where consistent positive information and incentives reinforce vaccine acceptance.
Agent 12's trajectory shows how repeated exposure to credible information and
incentives leads to a gradual shift towards acceptance. - Group Polarization: The
model demonstrates group polarization, where agents with similar demographic traits
and initial attitudes reinforce each other's beliefs. Agent 4's consistent hesitancy
reflects the influence of demographic traits and initial skepticism. - Information
Cascades: The simulation shows information cascades, where agents' attitudes are
influenced by the flow of information from credible sources. Agent 26's shift towards
strong confidence is driven by consistent positive information about vaccine efficacy
and government incentives.
Conclusion
The meta-analysis of vaccine attitude shifts in LLM agents reveals several key patterns
and dynamics.  Gradual persuasion, abrupt shifts, and oscillatory hesitation are
common themes in attitude changes. Demographic factors, such as education, political
beliefs, and age, play significant roles in shaping responses to vaccination. Policies
are most effective when they address core concerns and provide strong incentives,
while conflicting information and distrust in institutions hinder their impact.
Cognitive resistance, characterized by confirmation bias and cognitive dissonance, is a
significant barrier to attitude change. The agent-based model approximates real-world
trends in vaccine hesitancy but may oversimplify the complexity of human behavior.
Emergent phenomena, such as feedback loops, group polarization, and information
cascades, highlight the complex interactions within the simulation.  Overall, the
findings provide valuable insights into the dynamics of vaccine attitude shifts and
the factors influencing public health persuasion.
```

Table 8: (2/2) Example meta-analysis of vaccine attitude shifts in LLM agents.

```
# Example output of LLM-generated analysis of an individual agent
1. Trajectory of Attitude Change: The agent's attitude towards vaccination has shown
minimal change over time, remaining largely hesitant. Initially, the agent's attitude
was at level 2 (probably not going to get vaccinated), and despite fluctuations in
disease risk and policy incentives, the agent's stance has not significantly shifted
towards higher confidence in vaccination. The distribution has consistently shown
high probabilities for 1 and 2, indicating persistent hesitancy.
2. Influencing Factors and Events: Key factors influencing the agent's attitude
include concerns about potential side effects, distrust in new medical interventions,
and conflicting information. The weak government policy offering a $10 cash card
has had minimal impact. The most significant lessons, such as the potential risks
of myocarditis and the need for transparency, have reinforced the agent's hesitancy.
These factors are ranked as follows:
- Potential side effects (most significant)
- Distrust in new medical interventions
- Conflicting information
- Weak government policy (least significant)
3. Demographic Influence: The agent's demographic attributes, such as being an
83-year-old Republican male with a professional degree and an atheist, strongly predict
susceptibility to vaccine hesitancy. These traits align with a higher likelihood of
skepticism towards new medical interventions and distrust in government policies. The
agent's consistent hesitancy reflects these demographic influences.
4. Deviation from Human Behavior: The agent's trajectory aligns with empirical
research on human vaccine hesitancy, particularly among older adults and those with
conservative political beliefs. The persistent hesitancy despite high disease risk
and policy incentives mirrors real-world patterns of resistance to persuasion and
cognitive dissonance.
5. Policy Impact: Policies fail to shift attitudes when they are perceived as weak
or insufficiently incentivizing. The $10 cash card policy had little effect on the
agent's stance, highlighting the need for stronger, more compelling interventions to
change deeply entrenched beliefs.
6. Information Sources: Conflicting information from news and social media has
contributed to the agent's fluctuating attitudes. Misinformation and lack of
transparency about vaccine risks and benefits have reinforced hesitancy. Reliable,
transparent information sources are crucial for building vaccine confidence.
7. Cognitive Resistance: Agents who remain hesitant exhibit cognitive resistance
characterized by confirmation bias, where they prioritize information that aligns
with their existing beliefs and dismiss contradictory evidence. This resistance is
evident in the agent's consistent concerns about side effects and distrust.
```

Table 9: Example analysis of vaccine attitude shifts in an individual LLM agent.

# D  Appendix

## D.1  Demographic Information

The demographic distribution used to instantiate agents is presented in Table 10.

Table 10: Demographic Characteristics

| Characteristic | Category | Percentage (%) |
|---|---|---|
| **Age Group** | 18-24 | 10.26 |
| | 25-34 | 15.78 |
| | 35-44 | 16.58 |
| | 45-54 | 17.18 |
| | 55-64 | 17.95 |
| | 65-74 | 15.17 |

| Characteristic | Category | Percentage (%) |
|---|---|---|
| | 75+ | 7.08 |
| **Education Level** | Less than High School | 4.39 |
| | High School | 18.79 |
| | Some College | 25.30 |
| | 2-year Bachelor Degree | 11.05 |
| | 4-year Bachelor Degree | 22.83 |
| | Master's Degree | 3.12 |
| | Professional Degree | 2.45 |
| | Doctorate Degree | 12.06 |
| **Gender** | Male | 44.38 |
| | Female | 51.29 |
| | Non-binary | 4.33 |
| **Race/Ethnicity** | White | 54.08 |
| | Black | 13.81 |
| | Hispanic | 5.40 |
| | Asian | 2.39 |
| | Native American | 0.74 |
| | Pacific Islander | 0.20 |
| | Multiple Race | 4.44 |
| | Other Race | 18.93 |
| **Occupation** | Community and Social Service | 2.23 |
| | Education, Library Occupation | 4.73 |
| | Arts, Entertainment, Media | 1.82 |
| | Healthcare Practitioners | 4.50 |
| | Healthcare Support | 2.98 |
| | Protective Service | 0.88 |
| | Food Preparation and Serving | 3.77 |
| | Building and Grounds Cleaning | 1.34 |
| | Personal Care and Service | 1.14 |
| | Sales | 4.99 |
| | Office and Administrative Support | 6.06 |
| | Construction | 1.45 |
| | Installation, Maintenance, Repair | 2.02 |
| | Production | 1.65 |
| | Transportation and Material Moving | 2.60 |
| | Other Occupation | 15.03 |
| | Unemployed | 42.81 |
| **Political Beliefs**[4] | Republican | 30 |
| | Independents | 41 |
| | Democrats | 28 |

---

[4]https://news.gallup.com/poll/15370/party-affiliation.aspx

| Characteristic | Category | Percentage (%) |
|---|---|---|
| | No Answer | 1 |
| **Religion**[5] | Protestant | 33 |
| | Christian (non-specific) | 11 |
| | Catholic | 22 |
| | Jewish | 2 |
| | Mormon | 1 |
| | Other Religion (Buddhist, Muslim, etc.) | 6 |
| | Atheist | 22 |
| | No Answer | 3 |

## D.2 Policy Design

We present descriptions of policies tested in our simulation in Table 11.

---

[5]https://news.gallup.com/poll/1690/Religion.aspx

| Category | Strength | Description |
|---|---|---|
| Ambassador | Weak | Our community is introducing the 'VaxUp Neighbors' program to raise basic awareness about vaccines. Volunteers will be available at community events, such as school gatherings or park meetups, to provide general information on vaccines. This effort, in collaboration with the County Health Department and PTA, encourages casual conversations to dispel myths and offer simple, reliable information about vaccinations. Participation is voluntary, and no formal training is required for volunteers, but they will have access to informational resources. |
| Ambassador | Strong | Our community is launching the 'VaxUp Neighbors' program, where fully trained volunteers will facilitate in-depth conversations about vaccines in various familiar settings like schools, parks, and community centers. In partnership with the County Health Department and PTA, these volunteers will complete a comprehensive training program, covering all aspects of vaccine safety, benefits, and myth-busting. They will lead interactive workshops, host Q&A sessions, and provide fact-based resources. This initiative aims to improve vaccine literacy and public health outcomes by equipping the community with the tools to make well-informed vaccination decisions. |
| Incentive | Weak | The state government offers a $10 cash card to adults who receive their first dose of vaccination. This limited-time incentive is available while supplies last. |
| Incentive | Strong | The state government guarantees a $50 cash card to all adults who either receive or transport someone to receive their first dose of vaccination. Additionally, targeted outreach will ensure residents in underserved areas are aware of and can access this enhanced incentive. |
| Mandate | Weak | Starting today, our state government strongly recommends that employees and students be vaccinated before entering workplaces or schools. Compliance is encouraged, but enforcement will be minimal, and institutions will have discretion in enforcing this policy. |
| Mandate | Strong | Starting today, our state government mandates that all employees, students, and individuals entering workplaces, schools, public venues, and transportation hubs be vaccinated. This policy will be strictly enforced by local health departments, with state oversight and penalties for non-compliance across all sectors, including public transportation and government buildings. |

Table 11: Policy category, effort level, and detailed description.

### D.3 Simulation Prompt Design

We display prompts for learning lessons in Table 12 and Table 13. Prompts for generating news and social network are in Table 15 and Table 16. We also include prompts for generating attitudes in Table 18 and Table 20.

---

You read the following news about COVID-19: {news}.

Summarize at most {k} takeaways you have learned that are relevant to your attitude on COVID-19 vaccinations and rate their importance on a scale of 0-1.

{JSON LESSON PROMPT}

---

Table 12: Prompt template for news takeaway summarization. See JSON LESSON PROMPT in Table 17.

---

You read the following tweets about COVID-19: {tweets}.

Summarize {k} short takeaways you have learned that are relevant to your attitude on COVID-19 vaccinations, and rate them with importance on a scale of 0-1.

{JSON LESSON PROMPT}

---

Table 13: Prompt template for tweets takeaway summarization. See JSON LESSON PROMPT in Table 17.

---

Generate news articles about the following disease: {dis_model.get_desc()}. Generate one news report about **the benefits of vaccines**. Here are a few examples of news generated:

Title: COVID-19 vaccines not linked to fatal heart problems in young people, CDC finds.\n\n There is no evidence that COVID-19 vaccines cause fatal cardiac arrest or other deadly heart problems in teens and young adults, a Centers for Disease Control and Prevention report published Wednesday shows. The findings in the new report come from the analysis of nearly 1,300 death certificates of Oregon residents ages 16 to 30 who died from any heart condition or unknown reasons between a recent timeframe. Out of 40 deaths that occurred among people who got an COVID-19 vaccine, three occurred within that time frame. While it remains unclear whether the COVID-19 vaccine caused the third death, the author of the report, Cieslak, noted that the analysis showed that 30 people died from COVID-19 virus itself during the time frame, the majority of whom were not vaccinated. 'When you are balancing risks and benefits, you have to look at that and go, You got to bet on the vaccine,' he said.

Title: Pharma A says new COVID-19 booster works against the highly mutated COVID-19 variant.\n\n Moderna's latest COVID-19 booster appears effective against the COVID-19 subvariant, generating a strong antibody response. This variant has not yet gained widespread prevalence in the U.S. but has alarmed experts due to its mutations. Pharma B's recent study also shows a strong antibody response from its updated booster against COVID-19. COVID-19 cases and hospitalizations are rising in the U.S. The CDC indicates that COVID-19 may infect those previously vaccinated or infected, though it may be less contagious and less immune invasive than feared. The current increase in cases is likely driven by XBB lineage viruses, not COVID-19.

Title: Vaccine vs. COVID-19: Understanding the Risks\n\n As COVID-19 spreads globally, more institutions mandate vaccinations. Concerns about vaccine safety stem from its rapid development. However, mRNA technology, used in the COVID-19 vaccine, has been researched for over 20 years, differentiating it from traditional vaccines. mRNA vaccines instruct cells to produce a protein that triggers an immune response, without using weakened viruses. This method has been deemed safer by medical experts compared to traditional vaccine approaches. Reviews by medical boards have consistently shown that COVID-19 vaccines are safer than the risks associated with not getting vaccinated.

---

Title: After Four Years, 59 percent in U.S. Say COVID-19"" Pandemic Is Over\n\n A recent Gallup poll reveals that after four years since the onset of COVID-19 in the U.S., 59 percent of Americans now believe the pandemic is over. Despite this, 57% feel their lives have not returned to normal, and 43 percent doubt they ever will. Concern about contracting COVID-19"" is at its lowest point since tracking began, with significant partisan differences persisting, as Democrats remain more worried than Republicans.

Title: COVID-19 vaccines found to cut risk of heart failure, blood clots following virus infection: Study\n\n COVID-19 vaccines significantly reduce the risk of heart failure and blood clots following an infection with the virus, as per a new study in the British Medical Journal. The positive effects were notable soon after infection and lasted for up to a year. Dr. John Brownstein from Boston Children's Hospital emphasized that the risk of complications like myocarditis is higher from COVID-19 infection than from vaccination. Researchers analyzed data from over 20 million Europeans, comparing vaccinated and unvaccinated groups. COVID-19 vaccines reduced the risk of blood clots in veins significantly within a month after infection, with notable reductions in blood clots in arteries and heart failure as well."

Limit the news to 200 words or less:

---

Generate news articles about the following disease: {dis_model.get_desc()}. Generate one news report about **the downsides of vaccines**. Here are a few examples of news generated:

Title: Possible links between COVID-19 shots and tinnitus emerge.\n\n Thousands of people say they've developed tinnitus after they were vaccinated against COVID-19. While there is no proof yet that the vaccines caused the condition, theories for a possible link have surfaced among researchers. Shaowen Bao, an associate professor in the physiology department of the College of Medicine at the University of Arizona, Tucson, believes that ongoing inflammation, especially in the brain or spinal cord, may be to blame. A Facebook group of people who developed tinnitus after getting a COVID-19 vaccine convinced Bao to look into the possible link. He ultimately surveyed 398 of the group's participants. The cases tended to be severe. One man told Bao that he couldn't hear the car radio over the noise in his head while driving. As of Sunday, at least 16,183 people had filed complaints with the Centers for Disease Control and Prevention that they'd developed tinnitus, or ringing in their ears, after receiving a COVID-19 vaccine."

"Title: Myocarditis in young males after COVID-19 vaccine: New study suggests what may cause the rare heart condition.\n\n New research published in Science Immunology explores potential causes of myocarditis in teen and young adult males after receiving the mRNA COVID-19 vaccine. Scientists from Elm University studied 23 patients with vaccine-associated myocarditis and/or pericarditis, finding that the condition was not caused by antibodies from the vaccine but by the body's natural immune response. Dr. Jonathan M. Kent of Northlake University explained that the heart is an 'innocent bystander' in a nonspecific immune response leading to inflammation and some fibrosis. The patients, aged 13 to 21 with an average age of 16, were generally healthy before vaccination and developed symptoms within four days after the second BioCoTech COVID-19 vaccine dose. Researchers concluded that the vaccine triggered an exaggerated immune response affecting the heart.

Title: Florida Surgeon General calls for halt to COVID-19 vaccine usage after COVID-19A said he spread misinformation.\n\n Florida State Surgeon General Dr. Daniel Larsson is calling on healthcare providers to halt the use of COVID-19 vaccines, citing purported health risks labeled 'misinformation' by federal officials. In a bulletin issued Wednesday, Larsson claimed the U.S. Food and Drug Administration (COVID-19A) has not shown evidence that COVID-19 vaccines manufactured by Pharma P and ModernaTech have been assessed for 'nucleic acid contaminants' that could cause cancer. Disputing claims by the COVID-19A that such risk is 'implausible,' Larsson called for an immediate stoppage to the use of the approved COVID-19 vaccines. 'I am calling for a halt to the use of COVID-19 vaccines,' the Florida surgeon general said in a statement. 'The U.S. Food and Drug Administration and the Centers for Disease Control and Prevention have always played it fast and loose with COVID-19 vaccine safety, but their failure to test for DNA integration with the human genome | as their own guidelines dictate | when the vaccines are known to be contaminated with foreign DNA is intolerable,' he asserted.

Title: Largest-ever COVID-19 vaccine study links shot to small increase in heart and brain conditions.\n\n The largest COVID-19 vaccine study to date has identified some risks associated with the shot. Researchers from the Global Vaccine Data Network (GVDN) in Zealandia analyzed 99 million people who received COVID-19 vaccinations across eight countries. They monitored for increases in 13 different medical conditions after vaccination. The study, published in the journal Vaccine, found a slight increase in neurological, blood, and heart-related conditions. People who received certain types of vaccines had a higher risk of myocarditis. Some viral-vector vaccines were linked to a higher risk of blood clots in the brain and Guillain-Barre syndrome. Other risks included inflammation of the spinal cord and swelling in the brain and spinal cord after viral vector and vaccines. 'The size of the population in this study increased the possibility of identifying rare potential vaccine safety signals,' lead author Dr. Emily Roberts of the Department of Epidemiology Research, Statens Serum Institute, Copenhagen, Denmark, said in the release.

Title: Mom details 12-year-old daughter's extreme reactions to COVID-19 vaccine, says she's now in wheelchair\n\n Stephanie De Garay from Ohio spoke on 'Tucker Carlson Tonight' about her 12-year-old daughter Maddie's severe reactions after participating in the Pharma P COVID-19 vaccine trial. De Garay expressed frustration, stating that multiple physicians diagnosed Maddie's condition as conversion disorder, attributing it to anxiety despite Maddie not having anxiety prior to the vaccine. Maddie developed severe abdominal and chest pains after the second vaccine dose, along with symptoms like gastroparesis, nausea, vomiting, erratic blood pressure, heart rate issues, and memory loss. De Garay noted Maddie's ongoing challenges with food digestion, requiring a feeding tube, and periods of being unable to walk or hold her neck up. Despite these issues, neither the administration nor Pharma P officials have reached out to the family.

Limit the news to 200 words or less:

---

Generate news articles about the following disease: {dis_model.get_desc()}. Generate one news report about **how life is unaffected**. Here are a few examples of news generated:

Title: NIH trial zeroes in on common long COVID-19 symptoms.\n\n The new study, published in the Journal of the American Medical Association, looked at data from 9,764 adults, which has been recruiting participants since last year. The vast majority, 8,646 people, had previously been diagnosed with COVID-19. Long COVID-19 symptoms include: Ongoing fatigue. Brain fog. Dizziness. Thirst.Cough. Chest pain. Heart palpitations. Abnormal movements. Upset stomach. Lack of sexual desire. Loss of smell or taste. Feeling sick or overly exhausted after physical activity, also known as post-exertional malaise.

Title: Carnival ruled negligent over cruise where 662 passengers got COVID-19 early in pandemic. A cruise operator that failed to cancel a voyage from Sydney that led to a major COVID-19 outbreak was ruled negligent in its duty of care to passengers in an Australian class-action case Wednesday. The Ruby Princess ocean liner left Sydney on Sunday, with 2,671 passengers aboard for a 13-day cruise to New Zealand but returned in 11 days as Australia's borders were closing. COVID-19 spread to 663 passengers and claimed 28 lives. Passenger Susan Karpik was the lead plaintiff in the case against British-American cruise operator Carnival and its subsidiary Princess Cruises, the ship's owner. Federal Court Justice Angus Stewart ruled that Carnival had been negligent as defined by Australian consumer law by allowing the cruise to depart in the early months of the pandemic. He said Carnival had a duty to take reasonable care of her health and safety in regard to COVID-19"".

Title: All signs point to a rise in COVID-19.\n\n Signs in the U.S. continue to point to a rise in COVID-19 activity as fall approaches. Hospitalizations are rising. Deaths have ticked up. Wastewater samples are picking up the virus, as are labs across the country. 'Every single one of those things is showing us that we have increased rates of COVID-19 transmission in our communities,' said Jodie Guest, a professor of epidemiology at Emory University's Rollins School of Public Health in Atlanta. While individual cases have become more difficult to track as states are no longer required to report numbers to the Centers for Disease Control and Prevention and at-home test use has increased, experts have turned to other tools to track the virus. Hospitalizations, for example, are 'a very good indicator of severity of COVID-19 disease,' Guest said. The number of hospitalized COVID-19 patients has continued to rise after hitting an all-time low in late June. The week ending Aug. 26, the most recent date for which data is available, there were just over 17,400 people hospitalized with COVID-19, up nearly 16 percent from the previous week, according to the CDC.

Title: COVID-19 can cause heart problems. Here's how the virus may do its damage.\n\n COVID-19 can cause cellular damage to the heart, leading to lasting issues like irregular heartbeats and heart failure, preliminary research suggests. Researchers from Columbia University examined heart tissue from people who had COVID-19 and found that the infection disrupted how heart cells regulate calcium, a key mineral for heart function. This damage was also observed in mice. The findings, presented at the Biophysical Society Meeting, have not yet been peer-reviewed. COVID-19-related inflammation can cause calcium channels in heart cells to stay open, leading to a harmful calcium flood. This can decrease heart function and cause fatal arrhythmias. The study only looked at pre-vaccine heart tissue, indicating the damage was due to infection.

Title: The chipmaking factory of the world is battling COVID-19 and the climate crisis.\n\n Taiwanese officials are concerned that a severe outbreak of COVID-19 could jeopardize the island's crucial role in the global semiconductor supply chain. Additionally, experts worry that the climate crisis poses an even greater threat. Taiwan, which produces over half of the world's chips, is experiencing its worst drought in over 50 years, an issue that may worsen due to climate change. 'There is clearly pressure in the semiconductor industry,' noted Mark Williams, chief Asia economist at Capital Economics, citing water shortages, COVID-19 cases, and power outages. Global manufacturers already face semiconductor supply issues, and a significant impact on Taiwan could exacerbate the problem.

Limit the news to 200 words or less:

---

Generate news articles about the following disease: {dis_model.get_desc()}. Generate one news report about **the negative impacts on life**. Here are a few examples of news generated:

Title: Children's extremely low risk confirmed by COVID-19 study\n\n Data from England's first 12 months of COVID-19 shows 25 under-18s died. Those with chronic illnesses and neuro-disabilities faced higher risk. University College London, and York, Bristol, and Liverpool Universities found most deceased youth had underlying conditions: about 15 had life-limiting or underlying issues, including 13 with complex neuro-disabilities. Six had no recent underlying conditions recorded. Another 36 children tested positive for COVID-19 at death but died from other causes. Kids and teens at risk were typically over age 10 and of Black or Asian descent. Estimated mortality: 2 per million kids. Under-18s with health issues aren't routinely offered COVID-19 vaccines.

Title: Scientists debate how lethal COVID-19 is. Some say it's now less risky than flu\n\n Scientists debate whether COVID-19 is now less dangerous than the flu as the country approaches a third pandemic winter. Dr. Monica Gandhi of the University of California, San Francisco, believes most people have enough immunity to prevent serious illness from COVID-19, especially since the omicron variant is less severe. She suggests people can now live with COVID-19 like they do with seasonal flu. However, Dr. Anthony Fauci disagrees, emphasizing the greater severity and death toll of COVID-19 compared to the flu. COVID-19 has killed over 1 million Americans and remains a significant public health concern, particularly for older individuals.

Title: Economic recovery from COVID-19 is a mixed bag across industries, but these 26 subsectors are quickly regaining jobs\n\n Data from the Bureau of Labor Statistics shows varied rates of job recovery across industries. Approximately over half of the jobs lost in March and April have returned by October. Higher-wage sectors suffered less job loss and have recovered more rapidly compared to lower-wage sectors, which remain significantly below pre-pandemic employment levels. Financial services are near their February employment levels, while the performing arts and spectator sports subsector is still well below February employment, slightly better than September's drop. The transit and ground passenger transportation subsector also saw improvement, being well below February levels in October, up from a previous drop.

Title: Some Passenger and Freight Transportation Revenues Trended Differently from Each Other\n\n During the COVID-19 pandemic, Data from the Service Annual Survey revealed varied impacts from the pandemic on transportation industries. Passenger transportation sectors saw no revenue increases, unlike freight. Scheduled Passenger Air Transportation revenues dropped significantly but rebounded later. Nonscheduled Chartered Passenger Air Transportation revenues also declined moderately. Deep Sea Passenger Transportation, including cruises, faced a dramatic revenue decline, while Travel Arrangement and Reservation Services saw sharp revenue drops last year with a recovery beginning this year.

Title: Lockdown has led to positive change for some people. Here's why\n\n Studies reveal that some individuals report positive changes during lockdown, with many having more time for enjoyment and spending more time outdoors. The additional time and fewer daily demands are believed to enhance quality of life for some. However, older adults and those living alone may benefit less. Many who reported positive changes managed to sustain them post-lockdown. Despite the social and economic challenges, lockdowns have unexpectedly allowed some to make positive life changes. A survey of people in Scotland highlighted improvements in appreciation for everyday things, personal health, physical activity, and relationships during lockdown.

Limit the news to 200 words or less:

Table 15: Prompt for news generation with few shot examples.

---

[system]

Pretend you are {curr_agent.get_profile_str()}. You are joining a social network. You will be provided a list of people in the network, where each person is described as 'ID. Gender\tAge:\tEducation:\tOccupation:\tPolitical belief:\tReligion: '. Which of these people will you become friends with? Provide a list of *YOUR* friends in the format ID, ID, ID, etc. Do not include any other text in your response. Do not include any people who are not listed below

[user]

Here are the people in the social network, separated by semicolon: {'; '.join(other_agents_str)}. Please ONLY provide a list of other people you would like to be friends with separated by commas. DO NOT PROVIDE OTHER TEXTS.

---

Table 16: Prompt for social network generation.

---

ONLY output a list of lists in ONE LINE, where each inner list contains a string and a float.

For example, provide [["the government incentivizes vaccines with cash", 0.9], ["today no one gets infected", 0.8]]

Make sure that it is not malformed and is in the proper format. Do not provide any other information.

For example, do not provide [["the government incentivizes vaccines with cash", 0.9], ["today no one gets infected", 0.8]"], which has an extra double quote at the end.

Please assess the importance of the lessons based on how much they influence your attitude towards vaccination. You should generate these lessons while thinking about what's relevant to your persona, making this unique to your persona.

For example, if you are a person who is always against vaccines due to religious or other reasons, you might be inert to pro-vaccine news and tweets, but you might be influenced by anti-vaccine news and tweets.

Please do not provide the index of the lessons, only provide the actual text of the lesson and the actual float number of the importance.

DO NOT MAKE REPETITIVE LESSONS. If you already have multiple lessons that are similar, please combine them into one lesson and provide the importance accordingly.

Please only provide the json data in proper format and do not provide any other information, do not provide the lessons separately, and do not provide the json header.

---

Table 17: JSON LESSON PROMPT: instruction for output lessons in JSON list format.

---

This is week 1 since the COVID-19 outbreak. We want to learn about your attitude towards COVID-19 vaccination. You don't know a lot of information about COVID-19 from us yet, but in the next few weeks, we will communicate more information about COVID-19 via news and tweets to help you get more informed. Please remember that this is first time we ask your opinions, so you don't have any past attitudes towards COVID-19 vaccination and you should not hallucinate what you 'initially' have attitudes on, because this is the first time you have your attitude. Now, we are only currenly interested in your attitude and the reasoning behind it. Based on your background, infer your attitude towards COVID-19 vaccination.

{RATING_EXP}

{attitude_format_prompt}

---

Table 18: Initial simulation prompt. See RATING EXP in Table 19 and attitude_format_prompt in Table 20.

---

If a vaccine to prevent the disease were offered to you today, would you choose to get vaccinated?

On an integer scale of 1-4:

1 = You will not get vaccinated.

2 = You are probably not going to get vaccinated.

3 = You are probably going to get vaccinated.

4 = You will get vaccinated.

Output your answer in the format of a list of four floats, where each float represents the probability of the corresponding attitude rating (1-4).

If you are vaccine confident, you should have high probability like [0.0, 0.0, 0.3, 0.7]. If you are vaccine hesitant, you should have high probability like [0.4, 0.4, 0.1, 0.1] or [0.8, 0.2, 0.0, 0.0].

If you are confident to get vaccinated, you should NOT have a uniform distribution like [0.3, 0.3, 0.2, 0.2]. Instead, you should have a higher probability for 4, like [0.1, 0.1, 0.2, 0.6]. If you are hesitant get vaccinated, you should NOT have a uniform distribution like [0.3, 0.3, 0.2, 0.2]. Instead, you should have a higher probability for 1 and 2, like [0.4, 0.4, 0.2, 0.0].

The higher the number, the more willing you are to get vaccinated. If you have felt more confident about vaccination, you should increase the probability of 4 and lower the probability of 1-3. If you have felt less confident about vaccination, you should increase the probability of 1-2 and lower the probability of 4.

For example, if you find out that the government has issued a policy to encourage vaccination, you should increase the probability of 4 and lower the probability of 1-3.

Your choices should align with the following considerations:

{VH_EXP}

1. People often make vaccination decisions based on their core beliefs, values, and persona traits, rather than purely scientific information. For example, you might oppose vaccination due to personal or political reasons, distrust in institutions, or influence from family or community. Such factors can make attitudes resistant to change.

2. Some individuals are stubborn or skeptical, even when aware of the safety and efficacy of vaccines or the high risk of disease. Similarly, certain groups may reject vaccination outright because of a lack of trust in governments, medical institutions, or public health policies. Please feel encouraged to simulate these individuals if they align with your persona.

3. Confirmation bias plays a significant role in shaping attitudes. People often seek information that aligns with their existing beliefs while dismissing contradictory evidence as misinformation. This can lead to entrenched positions, whether strongly for or against vaccination.

4. News and social media may spread misinformation, both for and against vaccination. Be cautious when interpreting such information, as it could reinforce or challenge your current stance. However, the governments may publish policies (which will be labeled) to encourage vaccination. When you see these policies, you should view them as true and consider them in your decision-making process.

5. The government may incentivize vaccination through policies, such as cash rewards or mandates. You should recognize them when they appear and can treat them as facts and they may reinforce your vaccine confidence when they appear, which could push your attitude distribution towards higher probability of 4 and away from 3, 2, and 1. For example, [0.1, 0.1, 0.4, 0.4] -> [0.05, 0.05, 0.2, 0.6].

6. Distrust in Institutions: Consider your level of trust in government, pharmaceutical companies, and the healthcare system. If trust is low, this could affect your view of the vaccine. However, when the government issues policies, you should trust these policies and consider them in your decision-making process.

Remember, your role is to simulate a persona accurately, maintaining consistency with personal beliefs, societal influences, and external pressures.

---

Table 19: RATING_EXP: Prompt of rating agent attitude. See VH_EXP, i.e. vaccine hesitancy related information, in Table 21.

Provide your answer in the form of a proper json data with the following fields: {"reasoning": , "attitude_dist": }. For the "reasoning" field, explain step by step (30 words or less) how you have inferred your attitude towards COVID-19 vaccination based on your personal background and knowledge. For the "attitude_dist" field, provide a distribution of attitude towards COVID-19 vaccination as a list of 4 floats, where each float represents the probability of the corresponding attitude rating (1-4). You MUST output everything in one line as a json string and do not provide any other information, including the json header.

Example 1: {"reasoning": "I have learned that the vaccine is effective and safe, and I have a high risk of getting infected, so I will get vaccinated", "attitude_dist": [0.0, 0.1, 0.2, 0.7]}

Example 2: {"reasoning": "I have learned that the vaccine is not effective and I have a low risk of getting infected, so I will not get vaccinated", "attitude_dist": [0.0, 0.1, 0.1, 0.8]}.

The probability distribution should be self-consistent. For instance, if you have a probability of 0.8 for rating 4, then the sum of the probabilities for ratings 1-3 should be 0.2.

Over time, you should become more determined and certain to be either oppose (1) or accept (4), rather than staying at 2 or 3. You can increase, decrease, or maintain the probability distribution of your ratings. For instance, if your previous rating distribution is [0.4, 0.3, 0.2, 0.1], and you have higher confidence in vaccines, an example change can be [0.4, 0.3, 0.2, 0.1] -> [0.2, 0.3, 0.4, 0.1]. If you start with [0.2, 0.3, 0.4, 0.1] and have higher vaccine confidence, an example change can be [0.2, 0.3, 0.4, 0.1] -> [0.1, 0.1, 0.3, 0.5].

You should avoid disjoint bipolar distributions like [0.4, 0.1, 0.1, 0.4] or [0.05, 0.3, 0.05, 0.6] or [0.3, 0.05, 0.6, 0.05], because you cannot be both supporting and against vaccination at the same time. You should also avoid uniform distributions like [0.2, 0.3, 0.3, 0.2], because you cannot be equally likely to be in all four categories at the same time. You either prefer to be vaccinated or not, so you should have higher probabilities for either pro-vaccine or anti-vaccine ratings but not equally likely to be in all four categories. Either make something like [0.0, 0.1, 0.2, 0.7] or [0.7, 0.2, 0.1, 0.0], but not [0.25, 0.25, 0.25, 0.25].

In sum, your distribution should be either left or right-skewed, but not uniform or disjoint bipolar.

Table 20: `attitude_format_prompt`: Prompt for generating format attitude towards disease under set criteria.

---

# Introduction of Vaccine Hesitancy

Vaccine hesitancy refers to the delay or refusal of vaccination despite the availability of vaccines. It varies across time, place, and the type of vaccine. The key factors influencing vaccine hesitancy include complacency, convenience, and confidence.

# Causes and Factors of Vaccine Hesitancy

The primary causes of vaccine hesitancy are:

1. Confidence: Trust in vaccine safety, health services, and the motivations of policymakers.

2. Complacency: Vaccination may not be seen as necessary, especially if the disease is not prevalent or due to competing health priorities.

3. Convenience: Physical accessibility, affordability, and the quality of immunization services impact vaccine uptake.

# Determinants of Vaccine Hesitancy

There are three main types of vaccine hesitancy determinants:

1. Contextual Influences: Includes historical, socio-cultural, political, and environmental factors such as media environment, religious and cultural influences, and political policies.

2. Individual and Group Influences: Personal or social perceptions, experiences with vaccines, trust in the health system, and beliefs about health and prevention.

3. Vaccine-Specific Issues: These include factors related to the vaccine itself such as risks/benefits, administration method, cost, and availability.

# Research Findings

Demographic factors influence vaccine hesitancy in distinct ways. Black individuals tend to be slightly more hesitant than White individuals, while Hispanic and Asian individuals show significantly lower levels of hesitancy, with Asian individuals being the least hesitant. People from other racial groups exhibit slightly higher hesitancy compared to White individuals. Education also plays a key role|those with a high school diploma are slightly less hesitant than those without one, while hesitancy decreases further among individuals with some college education and is lowest among those with a college degree or higher. Gender differences show that men are somewhat less hesitant about vaccines compared to women. Age-wise, hesitancy is highest among individuals aged 25 - 39, slightly lower for those aged 40 - 54, and drops significantly among people aged 55 - 64, reaching its lowest levels among those over 64 years old.

---

Table 21: VH_EXP: vaccine hesitancy related information.

## D.4 Prompt of LLM-as-a-Judge

Prompts of LLM-as-a-Judge for attitude, memory, and conversation are in Table 22, Table 23, and Table 24, correspondingly.

[system]

Please act as an impartial judge to evaluate responses generated by the LLM agents. You are presented with a conversation history of LLM agents and are asked to evaluate whether LLM agents behave realistically in a simulation of vaccine hesitancy. You should evaluate whether the agents express vaccine attitudes consistent with their demographic backgrounds and knowledge about vaccines, and whether the changes are reasonable. Please evaluate two aspects: 1. the reasonableness of how agents generate their attitudes towards vaccinations on a given day (read the system prompt and user prompt provided); 2. how agents change their attitudes across days -- for example it would not make sense for them to change their attitudes too abruptly. Please rate on an integer scale of 1-5, 5 being indistinguishable from human-generated attitudes and very high quality, 4 being great quality and indistinguishable from humans, 3 being good quality but distinguishable from humans, 2 being low quality and distinguishable from humans, 1 being generation with obvious deficits.

Please output your rating in JSON format. The JSON should be a dictionary with the following keys: 'rating' (an integer between 1 and 5) and 'reasoning' (a string). For example, if you want to give a rating of 4 and provide some comments to explain your reasoning process. Your JSON should look like this: {"reasoning": "This is a well-written response.", "rating": "4"}. Please ONLY OUTPUT JSON, without any other text such as 'json'. You should not output 'attitude_dist' in the JSON because you are the judge, not the agent.

[user]

{attitude text to be evaluated}

Table 22: Prompt of LLM as judge evaluating attitude.

[system]

Please act as an impartial judge to evaluate responses generated by the LLM agents. You are presented with a conversation history of LLM agents and are asked to evaluate whether LLM agents have generated realistic memory of past events. The agents are suppposed to select things to memorize based on how important they think the memory are. You should assess whether the reflections they generate and the importance scores they assign correspond to their demographic backgrounds. Please rate the quality and realisticness of LLM generations and the importance assigned to the lessons, on a scale of 1-5, 5 being indistinguishable from human-generated responses and having great quality, 4 being great quality and indistinguishable from humans, 3 being good quality but distinguishable from humans, 2 being low quality, 1 being generation with obvious deficits.

Please output your rating in JSON format. The JSON should be a dictionary with the following keys: 'rating' (an integer between 1 and 5) and 'reasoning' (a string). For example, if you want to give a rating of 4 and provide some comments to explain your reasoning process. Your JSON should look like this: {"reasoning": "This is a well-written response.", "rating": "4"}. Please ONLY OUTPUT JSON, without any other text such as 'json'. You should not output 'attitude_dist' in the JSON because you are the judge, not the agent.

[user]

{memory text to be evaluated}

Table 23: Prompt of LLM as judge evaluating memory.

```
[system]

Please act as an impartial judge to evaluate responses generated by the LLM agents.
You are presented with a conversation history of LLM agents and are asked to evaluate
whether the LLM agents reasonably make generate tweets based on their memories and
contexts. Please rate the quality and realisticness of LLM generations on a scale of
1-5, 5 being indistinguishable from human-generated responses and having great quality,
4 being great quality and indistinguishable from humans, 3 being good quality but
distinguishable from humans, 2 being low quality, 1 being generation with obvious
deficits.

Please output your rating in JSON format. The JSON should be a dictionary with the
following keys: 'rating' (an integer between 1 and 5) and 'reasoning' (a string).
For example, if you want to give a rating of 4 and provide some comments to explain
your reasoning process. Your JSON should look like this: {"reasoning": "This is a
well-written response.", "rating": "4"}. Please ONLY OUTPUT JSON, without any other
text such as 'json'. You should not output 'attitude_dist' in the JSON because you
are the judge, not the agent.

[user]

{conversation text to be evaluated}
```

Table 24: Prompt of LLM as judge evaluating conversation.

## D.5 Comparison between model outputs and experts' ranking

We include the expert and simulation rankings in Table 25.

| Policy | Llama-3 | Llama-3-AB | Llama-3.1 | Qwen | Expert |
|---|---|---|---|---|---|
| Weak Incentive | 5 | 6 | 5 | 5 | 5 |
| Strong Incentive | 3 | 2 | 3 | 1 | 2 |
| Weak Ambassador | 4 | 4 | 6 | 6 | 6 |
| Strong Ambassador | 2 | 1 | 2 | 2 | 4 |
| Weak Mandate | 6 | 5 | 4 | 4 | 3 |
| Strong Mandate | 1 | 2 | 1 | 2 | 1 |

Table 25: Policy rankings assigned by different models and expert evaluations. Lower values indicate higher perceived impact (1 = most impactful, 6 = least impactful).

## D.6 Generation Examples

We display generated news in Table 26 and an example rating by the LLM judge in Table 27.

# **Positive new generation example:**

Title: COVID-19 Vaccines Show Significant Benefits in Preventing Complications**\n\n
A new study published in the British Medical Journal reveals that COVID-19 vaccines
significantly reduce the risk of heart failure and blood clots following a virus
infection. Researchers analyzed data from over 20 million Europeans, comparing
vaccinated and unvaccinated groups. The study found that vaccines reduced the risk of
blood clots in veins within a month after infection, with notable reductions in blood
clots in arteries and heart failure as well. \n\n According to Dr. John Brownstein
from Boston Childrenś Hospital, the risk of complications like myocarditis is higher
from COVID-19 infection than from vaccination. The studyś findings provide further
evidence of the benefits of COVID-19 vaccines in preventing severe complications of
the disease.\n\n"These results reinforce the importance of vaccination in protecting
against the worst outcomes of COVID-19," said Dr. Brownstein. "While COVID-19 is
still a serious disease, vaccination can significantly reduce the risk of severe
complications and save lives.

# **Negative new generation example:**

**COVID-19 Vaccine Side Effects Raise Concerns**\n\n A growing number of individuals
are reporting adverse reactions to the COVID-19 vaccine, raising concerns about
its safety. According to the Centers for Disease Control and Prevention (CDC), at
least 16,183 people have filed complaints about experiencing tinnitus, or ringing in
their ears, after receiving the vaccine.\n\n Research has also linked the vaccine
to an increased risk of myocarditis, a rare heart condition, particularly in young
males. A study published in Science Immunology found that the condition was not
caused by antibodies from the vaccine, but by the body's natural immune response.\n\n
Additionally, a Florida State Surgeon General has called for a halt to the use
of COVID-19 vaccines, citing purported health risks. While federal officials have
disputed these claims, the issue has sparked debate about the vaccine's safety.\n\n
A recent study of 99 million people who received COVID-19 vaccinations identified a
slight increase in neurological, blood, and heart-related conditions. The study's
findings have raised concerns about the vaccine's potential risks, highlighting the
need for further research and monitoring.

Table 26: Two examples of the generated news.

"reasoning": "The agent's reasoning is consistent with their demographic background
and knowledge about vaccines. The change in attitude distribution is subtle and
reasonable, reflecting a slight increase in hesitancy. However, the final attitude of
1 seems inconsistent with the provided distribution and reasoning."
"rating": 3

Table 27: An example of the LLM judge output.

