# OpenReview forum: "Can A Society of Generative Agents Simulate Human Behavior and Inform Public Health Policy? A Case Study on Vaccine Hesitancy"
_colmweb.org/COLM/2025/Conference — COLM 2025_

### Official Review · Reviewer_wsJL · 2025-05-11

**Rating:** 7
**Confidence:** 3
**Ethics Flag:** 1

**Summary:**

In this paper, the authors propose the VACSIM framework for simulating human behavior to inform public health policy using 100 carefully designed agents. To better reflect real social situations, the framework incorporates several components inspired by previous work, such as persona construction, memory and lesson storage, and vaccine attitude change probabilities based on prior surveys.

Moreover, the paper includes modules for news, social networks, and perceived disease risk, allowing agents to interact with their environment. The framework also considers the delay in vaccine policy implementation following a disease outbreak.

For evaluation, the authors use detailed metrics and various models. The findings span from global to local perspectives and include comparisons with real-world data and expert opinions, demonstrating that this simulation framework behaves similarly to real-world dynamics.

**Questions To Authors:**

- Could the authors elaborate on how this social simulation framework could be applied to other domains, such as misinformation, which the paper briefly mentions?
- Regarding the warmup phase, can you provide more details on other possible factors influencing the warmup settings or any additional findings not included in the paper?
- Since the code is not released, could you clarify which frameworks and libraries were used in the project?

**Reasons To Accept:**

- The authors comprehensively consider both individual agents and the overall system, including many components and detailed modeling of social and news network influences, referencing prior domain work to create more realistic scenarios.
- The warmup phase is particularly interesting, simulating government announcements and reaction times, which helps explain the results.
- The detailed prompts in the appendix are valuable and could be generalized to other tasks.
- The paper thoroughly explains different cases across settings, and the social findings provide meaningful insights for policymakers designing corresponding interventions.

**Reasons To Reject:**

- The total number of agents is limited; 100 is relatively small for large-scale social simulations.
- Due to the complexity of components, there are concerns about execution time when scaling up.
- The codebase is not publicly released.
- The paper misses related work such as https://arxiv.org/abs/2410.13915, which focuses on multi-agent simulations of misinformation using real social media platforms and is highly relevant.

---

> ### Author Response · Authors · 2025-06-03
> **Thank you for your review**
>
> Dear Reviewer wsJL:
>
> Thank you for your thorough and thoughtful review.
>
> ### **Scalability Analysis**
>
> We acknowledge that 100 agents is modest for large-scale social simulations. This design choice was driven by resource constraints, as LLM-powered agents are significantly more computationally intensive than rule-based agents. Nevertheless, as noted in our general response, we have run additional experiments (experiment #2) with varying numbers of agents and found that both simulation run-time and attitude dynamics scale roughly linearly. These results suggest the feasibility of scaling to larger agent populations with optimized infrastructure. To that end, we are also exploring frameworks like AgentTorch (Chopra et al., 2024), which enable scaling up to millions of agents to support future work at larger scales.
>
> ### **Repo and Libraries**
> We plan to release the codebase alongside the public version of the paper. The simulation uses a modular architecture built primarily with:
> - vLLM for serving local model servers and efficient batched inference across agents
> - Python multiprocessing for parallel simulation execution
> - OpenAI API for accessing local model servers
> - SentenceTransformers (Reimers & Gurevych, 2019) for embedding tweets/news and similarity computation.
> - We will ensure that the final release includes complete documentation and pretrained configurations for replicability.
>
> ### **Related Work**
> We are also very grateful for the related work you point out. It is indeed highly relevant to our paper and we will include this in our camera-ready version.
>
> ### **Applications to Other Domains**
> Thank you for your question and for encouraging us to consider broader applications of the framework. While our current implementation is focused on vaccine attitudes, the underlying simulation infrastructure is modular, and we believe it could be adapted with additional development for use in related domains such as **misinformation** and **disaster response**.
> ### **Misinformation**:
> We believe that adapting our framework to simulate health-related misinformation is a natural next step. The current components, such as agent-level variation in trust, social network structure, and exposure to news, can be extended to model how false or conflicting information propagates and influences decision-making. While we have not yet implemented such a domain, we can incorporate more targeted prompts and misinformation-specific dynamics, including credibility of sources and susceptibility profiles in future iterations. By varying message content, source credibility, and network structure (such as using trust data such as from the KFF Tracking Poll: https://www.kff.org/health-information-trust/poll-finding/kff-tracking-poll-on-health-information-and-trust-the-publics-views-on-measles-outbreaks-and-misinformation/) we can assess how these factors impact vaccination decisions and test interventions (e.g., fact-checking, trusted messengers, message framing).
>
> ### **Disaster Response**:
> Another potential direction is modeling behavior under crisis conditions, such as during wildfires or other natural disasters. We sketch an example adaptation below:
>
> *A wildfire event can be introduced as a time-evolving environmental stimulus. Agents receive alerts, updates, and rumors from various sources (government, social media, neighbors), simulating the information landscape during a crisis. Factors such as household size, mobility limitations (e.g., elderly or disabled residents), and previous disaster experience can influence agent decisions to evacuate or shelter in place. Agents interact with one another, shaping risk perception and behavior through social cues, peer reassurance, or panic, capturing emergent group-level patterns like compliance or congestion. Public safety policies (e.g., targeted alerts, phased evacuation orders, or hyperlocal outreach) can be tested and compared in terms of their effect on response timing and safety outcomes.*
>
> While we have not yet tested this setting, the simulation infrastructure offers a promising foundation for exploring how individuals react to uncertainty, peer influence, and institutional broadcasts during emergencies.
>
> We have added a discussion of these extensions in the conclusion to reflect both the **flexibility and current limitations** of our framework. We hope to explore these applications more fully in future work. Thank you again for highlighting this important direction.

---

> > ### Comment · Reviewer_wsJL · 2025-06-05
> >
> > Thank you for the concise explanation of the infrastructure components. The disaster response and misinformation aspects represent two compelling research directions, and I appreciate your detailed description of these scenarios. Incorporating these elements into the paper would significantly enhance the work's extensibility and improve its generalizability.
> > I will maintain my positive evaluation.

---

> > > ### Author Response · Authors · 2025-06-06
> > >
> > > Dear Reviewer wsJL,
> > >
> > > Thank you so much for your affirmation and prompt response!

---

### Official Review · Reviewer_eATj · 2025-05-12

**Rating:** 7
**Confidence:** 4
**Ethics Flag:** 1

**Summary:**

The paper introduces an agent-based model using an LLM to simulate the effects of public policy on vaccine heistance. To do this, the authors use a population of 100 agents, with unique personas. The simulation includes a social network, news, and policy announcements. The paper alos introduces attitude modulation and simulation warmup to making the simulations more realistic.

**Reasons To Accept:**

- The central contribution of the paper is quite unique. While agent-based models have been used for public policy simulation, using general LLMs as a central component has not been explored in as much detail.
- The paper is well written and easy to understand.
- The problem is well motivated.
- I really liked the overall setup of the world with the social network, the news, and public policy announcements.
- The breadth of models tested is comprehensive covering different model sizes and families and includes open source models. Further, the best results use 8 B/7 B models, making replication feasible.
- Systematic evaluation rubric of Verification (internal validity) and Verification (External validity) could become a standard for such simulations.
- It was good to see that the models that were good at global and local consistency had a higher correlation with human expert rankings.
- The qualitative analysis was also pretty insightful to read!

**Reasons To Reject:**

- The figures aren’t particularly clear. The authors could improve the presentation of the results.
- I would have liked to see the individual components of the simulation tested separately:
    - How good is the recommender system?
    - What is the quality of news generated
    - Some measure of inter-rater reliability of gpt-4o and human raters while checking the quality of responses. Some papers have shown that gpt-4o likes its own responses and longer response more.
- It is not clear to me how important each part of the simulation is. I would have liked to see some controls in the experiments: either ablations, or some forms of positive / negative controls.
- No confidence intervals for some results.

---

> ### Author Response · Authors · 2025-06-03
> **Thank you for your review**
>
> Dear Reviewer eATj,
>
> We are grateful for your detailed review.
>
> ### **Presentation Improvement**
>
> We appreciate your comments regarding the clarity of our figures. In the current version, Figure 4 aggregates multiple evaluations into a single visual to conserve space. In the camera-ready version, we will consider separating these sub-figures, improve axis labeling and legend clarity, and provide detailed captions. We will also include a compact summary figure/table to help readers understand the simulation pipeline in addition to the illustrative example in Figure 3.
>
> ### **Component-wise Analysis**
>
> We agree that a component-wise analysis would improve interpretability. In response, we have:
> - Conducted an ablation of the **fixed bias parameter** used in tweet recommendations. Results in the general response show that hesitancy trends are stable across the range [0.2, 0.4], suggesting moderate reliability of the social recommendation module.
> - Evaluated the **quality of generated news** by comparing GPT-generated articles with real-world news from the HuggingFace COVID News Dataset. We used GPT-4 to rate both sets blind to source on a scale of 1-5 (1 being poor and 5 being excellent) and control the news to have the same lengths. Results show that AI-generated news have average scores of 3.817 and 4.043, which demonstrates comparable quality.
>
> ### **Self-rating Bias**
>
> Thank you for highlighting the issue of LLM raters favoring their own outputs, a phenomenon that has been extensively documented in prior work (e.g., AlpacaEval). To mitigate this, in our local consistency evaluation (Section 4.3):
> - We do not use GPT-family models as both evaluator and generator. No GPT-series models are among those being evaluated in the P4 of Figure 4.
> - We also enforce length constraints on generated responses to control for verbosity bias in evaluation.
> We will clarify these methodological safeguards more explicitly in the final version.
>
> ### **Ablations**
>
> We appreciate the suggestion to include positive/negative controls or ablations to better isolate the contribution of each component. We have already conducted the following:
> - In Figure 4 (P1), we compare simulations with and without **attitude modulation**, showing improved real-world alignment for several models (e.g., Llama-3.1).
> - We run additional ablations of the **warm-up phase**. We remove the warm-up phase and perform 20-step simulations under the strong incentive policy for five different random seeds, and evaluate the MAE. It increases from 2.87% (with warm-up) to 5.00% (without warm-up). Without warm-up, the vaccine hesitancy reduction becomes more drastic with a less smooth curve, which shows that the warm-up phase is helpful for a more reliable simulation. We will include these results in the updated appendix in the camera-ready version.

---

> > ### Comment · Reviewer_eATj · 2025-06-03
> >
> > Thank you for your response!
> >
> > I think the new experiments improve the soundness of the paper! I will keep my initial positive score!

---

> > > ### Author Response · Authors · 2025-06-04
> > >
> > > Dear Reviewer eATj,
> > >
> > > We are sincerely grateful of your prompt response and acknowledgment!

---

### Official Review · Reviewer_2uyB · 2025-05-12

**Rating:** 8
**Confidence:** 4
**Ethics Flag:** 1

**Summary:**

The paper investigates whether generative agents can help policymakers trial public-health interventions. The authors build a simulated society of 100 LLM-driven agents. Each agent responds to a chosen incentive. Attitude-modulation and a warm-up period that lets opinions settle before any policy is applied are used. The work contributes the framework—including pipelines for agent creation, news generation, social-network construction and risk signalling—and introduces attitude-modulation plus simulation warm-up. Finally, it offers a systematic evaluation protocol that identifies which current LLMs are trustworthy for simulation.

**Questions To Authors:**

Thank you for this timely, insightful, and interesting submission. I enjoyed reading it and learned a lot from it. Some of the key limitations and areas for improvement include:

1. The paper arbitrarily fixes the simulated society at 100 agents. Phenomena such as opinion cascades, clustering, and complex contagion typically surface only in populations of thousands. A scalability study varying agent counts and measuring stability, run-time, and emergent dynamics—would clarify whether conclusions hold beyond this arbitrary setting.

2. Claims of being "pioneering" should be toned down. Prior work has already used agent-based and LLM-driven simulations to study vaccine uptake and other health behaviors. The true contribution is the specific combination of social-media inputs, and attitude modulation, not the idea of generative agents for public-health policy.

3. The biggest issue is the consideration of the models' apparent pro-vaccine bias, inherited from their pre-training or alignment guardrails. The authors note that two LLMs cannot reproduce realistic hesitancy, but don't consider or discuss why. For example, previous work has found that safety guardrails can skew LLM outputs, effectively encoding bag-of-world-like behaviors where expressing concerns about sensitive topics such as vaccination is not allowed even when meaningful in context or otherwise desirable. An ablation that removes alignment layers or fine-tunes on balanced pro-/anti-vaccine text could reveal whether the bias stems from system-level safety filters or deeper corpus skew. Without this, the recommendation that those models are "unsuitable" remains speculative.

**Reasons To Accept:**

1. The paper addresses an important topic
2. Comprehensive experiments
3. Interesting and timely insights

**Reasons To Reject:**

1. Somewhat arbitrary or not strongly justified design choices
2. Exaggerated novelty claims
3. Lack of empirical investigation into the discovered phenomena
(please see below for more details)

---

> ### Author Response · Authors · 2025-06-03
> **Thank you for your review**
>
> Dear Reviewer 2uyB,
>
> We appreciate your acknowledgement of our paper and your insightful feedback. We agree that a scalability study would be very interesting and clarifying. We run additional experiments with larger agent populations. Due to resource and time constraints during the discussion period, we ran on the scale of hundreds of agents. Please refer to our results and analyses in the general response (additional experiment 2), which show that the simulation behaves regularly on the scale of hundreds. We fully agree that emergent phenomena such as complex contagion, clustering, or opinion cascades require testing at larger scales. We are actively exploring frameworks like AgentTorch (Chopra et al., 2024), which offers infrastructure to scale multi-agent simulations to thousands or millions of agents. This is a key direction for future work, which we will address in our discussion of limitations.
>
> We also thank you for suggesting we should tone down our paper. We will modify the phrasing in the contribution sections (where claims “pioneering”) and focus on highlighting the social media input and attitude modulation, following your advice.
>
> We appreciate your concern about the potential pro-vaccine bias that may be inherited from LLM pretraining and the need for safety alignment. This is an important and underexplored issue. As a partial step, we did evaluate Llama-3-8B-abliterated, an uncensored model trained to bypass refusals to safety-related prompts (Arditi et al., 2024). As shown in Figure 4, its behavior does not substantially differ from the instructed version on vaccine hesitancy. We suspect that it is because Llama-3-8B-abliterated is trained on safety-related data, which does not include vaccine-specific data and does not debias pro-vaccine tendency.
>
> We believe the reviewer is absolutely right in highlighting the need for a deeper investigation. This is an active area of interest for us, and we will note this limitation more explicitly in the final version. In future work, we aim to finetune agents on individual-level, vaccine-related demographic data to better capture the diversity of vaccine opinions and reduce systemic bias in simulation outcomes.

---

> > ### Comment · Reviewer_2uyB · 2025-06-07
> >
> > Thank you for clarifying and incorporating additional experiments and discussions.

---

> > > ### Author Response · Authors · 2025-06-10
> > >
> > > Dear Reviewer 2uyB,
> > >
> > > Thank you for your acknowledgement and constructive feedback to our paper!

---

### Official Review · Reviewer_99J4 · 2025-05-14

**Rating:** 7
**Confidence:** 4
**Ethics Flag:** 1

**Summary:**

The paper introduces VACSIM, an agent-based simulation framework designed to model the evolution of vaccine attitudes within a society during a pandemic. VACSIM incorporates realistic elements, such as a diverse population of agents with individual personas, a simulated news network, a social network for agent interactions via "tweets," and a module reflecting perceived disease risk based on real-world data. The framework aims to simulate dynamics and assess the impact of various vaccine policies on public attitudes, providing insights for public health interventions. Despite its promising approach, the paper acknowledges that the framework is still in its early stages and requires further refinement.

**Questions To Authors:**

1. Could the authors elaborate on why the raw saliency score without normalization is less intuitive for the model's internal processing or subsequent calculations? What specific issues arise if the score is left unnormalized?

2. The paper presents an exponential function for attitude modulation. What is the theoretical justification for using the logarithmic probability in this form? How does this ensure the preservation of the original relative order of attitude preferences?

3. In the news recommendation system, the maximum cosine similarity is used. Have the authors considered other similarity measures or hybrid approaches that might better capture the relevance between news and past agent interactions? How might these alternatives impact the model's performance?

4. The tweet recommendation system includes a fixed positive bias (ϕ) for followed users. What was the rationale behind choosing a bias value of 0.3? Did the authors perform any sensitivity analysis to evaluate how different bias values impact the diffusion of information within the social network?

5. The end hesitancy is calculated as the average of the last 3-time steps. Why was this specific time frame chosen? How might the results change if a longer or shorter period were used for averaging?

6. The framework uses a warm-up stage of 5-time steps before policy intervention. What criteria were used to determine the duration of this warmup phase? How sensitive are the results to variations in the length of this initial period?

7. The paper frequently refers to appendices for key details on demographics, policies, and prompts. Could the authors include summaries or key elements of these sections in the main text to improve readability and clarity for the reader?

**Reasons To Accept:**

1. VACSIM provides an innovative method for simulating public health dynamics, especially related to vaccine attitudes during a pandemic, which is highly relevant in the current global context.

2. The model’s use of real-world data to simulate disease risk and the inclusion of diverse agent personas make the framework promising for generating actionable insights into vaccine attitudes and the effectiveness of public health policies.

3. The framework’s ability to model the complex interactions between social influence, perceived risk, and policy interventions holds significant potential for informing public health strategies and interventions.

**Reasons To Reject:**

1. Some aspects of the framework lack strong theoretical justification, such as the exponential function used for attitude modulation and the normalization of the saliency score. These points need further explanation to understand their impact on model behavior and results.

2. The framework heavily relies on certain parameters (e.g., cosine similarity for news recommendations and a fixed bias for followed users). The paper does not explore alternative methods or sensitivities to these choices, which could limit the model’s generalizability and robustness.

3. The paper mentions several parameters, such as the number of time steps for averaging hesitancy scores and the length of the warm-up phase, but does not explore their sensitivity in depth. This leaves questions about the framework's robustness across different configurations.

4. The paper frequently references appendices for important details on demographics, policy descriptions, and prompts, which can hinder readability and understanding. Key details should be summarized or integrated into the main text for better clarity and accessibility.

5. The model lacks empirical validation, with no real-world testing or comparison against actual data on vaccine attitudes. This makes it hard to assess the framework’s accuracy and real-world applicability.

---

> ### Author Response · Authors · 2025-06-03
> **Thank you for your review**
>
> Dear Reviewer 99J4,
>
> We genuinely thank you for your comprehensive consideration and thorough reviews.
>
> ### **Normalization of Saliency Scores**
>
> The saliency score (Eq. 1) is designed to rank memory lessons based on their relevance to agents, combining lesson importance and time decay (following popular work like Park et al., 2023). We normalize these scores to the [0,1] interval so that the saliency of older lessons remains comparable to newly assigned importance scores. Without normalization, older lessons with high absolute scores might dominate agent memory, even if their relative importance has decayed. This could skew the retrieval process and interfere with consistent memory-based reasoning. Normalization thus helps maintain relative comparability across time, consistent with prior work.
>
> ### **Order-preserving Attitude Modulation**
> For attitude modulation, we apply temperature scaling to the agent’s output distribution over vaccine attitudes. This method is widely used in machine learning (Hinton et al., 2015; Holtzman et al., 2019) to adjust distribution sharpness while **preserving the original order of preference** (when temperature > 0), as proven in Rahimi et al., 2020.
>
> ### **Alternative Similarity Metrics**
> We selected cosine similarity for news recommendation as it is simple, widely used, computationally efficient, and interpretable. Given the complexity of the overall simulation, we prioritized the simplicity of individual components. We agree that alternative similarity metrics (e.g., BERTScore, Jaccard similarity, etc.) are worth exploring, and we consider this a promising direction for future work.
>
> ### **Following Bias in Tweet Recommendation**
> For the following bias in tweet recommendations, we included a sensitivity analysis in the general response. As shown there, the system is insensitive across a range of values [0.2,0.25,0.3,0.35,0.4] with minimal difference on MAE, which supports that our system is not sensitive to perturbations in the following bias.
>
> ### **Average End Hesitancy**
> We compute end hesitancy as the average over the last 3 simulation steps to smooth high-frequency fluctuations at the end of the trajectory. Choosing a longer window (e.g., 5 steps) would represent 16–25% of our total simulation length (typically 20–30 steps), which would inaccurately represent terminal behaviors. In longer simulations (e.g., 50+ steps), a wider averaging window may be more appropriate, and we are interested in exploring variations in future work.
>
> ### **Warmup stage selection**
>
> The 5-step warm-up phase was selected based on preliminary runs without policy interventions. We observed that agent attitudes stabilize after approximately 5 steps. Extending this phase would increase computational cost without significantly improving convergence quality; hence, our choice represents a trade-off between stability and efficiency.
>
> ### **Empirical validations**
> Regarding empirical validations, while we acknowledge in our abstract and conclusion that VacSim is a first step and not intended for immediate policy deployment, we have incorporated **multiple empirical validation strategies** in the paper:
> - **Figure 5** shows that simulated trajectories can closely match real-world hesitancy trends (lowest MAE approximately 2.8%).
> - **Table** 1 demonstrates that policy rankings generated by Llama-3.1 simulation correlate strongly with expert survey rankings (Kendall's τ=0.733, p = 0.056).
>
> These results suggest the framework has **meaningful application potential**, which we emphasize as a foundational step for future deployment.
>
> ### **Readability**
>
> We appreciate the reviewer’s concern and agree that clarity is paramount. In the final draft, we will move key information, such as demographic schema, policy definitions, and prompt structures, from the appendix into the main text. In addition to Figure 3, which currently provides a detailed end-to-end example with persona, news, policy, and tweet generation, we will also add a compact summary table in the main text for reference.
>
>
> **Bib**:
>
> Ari Holtzman, Jan Buys, Li Du, Maxwell Forbes, Yejin Choi. 2019. The Curious Case of Neural Text Degeneration. https://arxiv.org/abs/1904.09751
>
> Amir Rahimi, Amirreza Shaban, Ching-An Cheng, Richard Hartley, Byron Boots. 2020. Intra Order-Preserving Functions for Calibration of Multi-Class Neural Networks. https://arxiv.org/pdf/2003.06820
>
> Hinton, Geoffrey, Vinyals, Oriol, and Dean, Jeff. 2015. Distilling the knowledge in a neural network. https://arxiv.org/abs/1503.02531.

---

> > ### Comment · Reviewer_99J4 · 2025-06-10
> >
> > Thank you for your response and the new experimental results—they strengthen the overall quality of the paper. I’ve updated my score to 7 accordingly.

---

> > > ### Author Response · Authors · 2025-06-10
> > >
> > > Dear Reviewer 99J4,
> > >
> > > Thank you so much for raising the score and acknowledgement of our paper!

---

### Author Response · Authors · 2025-06-03
**General Response #1**

We sincerely thank all reviewers for their detailed and constructive feedback. Given that several reviewers requested additional ablation experiments, we include the following results and analyses to address these points directly.

### **Additional Experiment 1: Varying Following Bias**
We conducted analysis on varying the bias term that was introduced to increase its visibility of social media recommendation (see Eq. 4 in Section 3.2). This bias term was originally set to 0.3. To suppress the variability of each run, we report the average results across, we run 20-step simulations under the strong incentive policy across 5 different random seeds. The results below report mean absolute error (MAE) relative to real-world data. (Note: MAEs may differ from Figure 5, which used 30 steps; we run and report 20 steps for all experiments here due to time constraints of the discussion period.)

| Following Bias | MAE   |
|----------------|-------|
| 0.2            | 4.60% |
| 0.25           | 3.67% |
| 0.3            | 2.87% |
| 0.35           |  4.67%     |
| 0.4            | 4.87% |

The results suggest that the simulation is not sensitive to a range of the following bias values, with 0.3 yielding the lowest MAE.

### **Additional Experiment 2: Scalability of Simulation**
In this experiment, we vary the number of agents in the simulation and run 20-step simulations under the strong incentive policy over 5 different random seeds. We report the average runtime per 20 steps in hours and their respective MAEs. Results show that the simulation run-time scales approximately linearly, and the MAEs remain comparable. Although the asymptotic property of the simulation is worthy of studying, due to the time constraints of the discussion period and practical compute constraints, we leave testing scales beyond 500 agents to future work.

| Number of agents | Average runtime per 20 steps (in hours) | MAE |
|------------------|----------------------------------------|---------|
| 100              | 0.897                                  | 2.87%
| 200              | 1.967                                  | 3.93%
| 300              | 3.500                                  | 3.96%
| 500              | 8.492                                  | 2.67%

---

### Decision · Program_Chairs · 2025-07-08

**Decision:**

Accept

**Comment:**

This paper makes a timely contribution at the intersection of AI and health policy, introducing a framework for simulating vaccine hesitancy dynamics and testing different policy interventions. Following other recent work on "societies of generative agents", these simulations use populations LLM-based agents with demographically-grounded personas that interact with news media and with each other via social media. The paper also makes a few technical innovations, including "attitude modulation" to soften LLM biases and a "simulation warmup" to ensure realistic initial conditions for the policy interventions. The authors' transparently discuss limitations (e.g. regarding LLM biases and scalability) and provided thorough response to reviewer concerns (including new experiments addressing parameter sensitivity and scalability that strengthen the paper). These could be further strengthened by additional ablation experiments (e.g. isolated testing of the recommender system, news generation quality, or the importance of other components, as suggested by eATj). However, the unanimous support from reviewers (three 7s and an 8), combined with the authors' constructive rebuttal, makes this a clear accept that is likely to inspire follow-up work.